# The enzymatic activity of inositol hexakisphosphate kinase controls circulating phosphate in mammals

Yusuke Moritoh [1✉], Shin-ichi Abe [1], Hiroki Akiyama [1], Akihiro Kobayashi [1], Ryokichi Koyama[1], Ryoma Hara [1], Shizuo Kasai [1] & Masanori Watanabe[1]

Circulating phosphate levels are tightly controlled within a narrow range in mammals. By using a novel small-molecule inhibitor, we show that the enzymatic activity of inositol hexakisphosphate kinases (IP6K) is essential for phosphate regulation in vivo. IP6K inhibition suppressed *XPR1*, a phosphate exporter, thereby decreasing cellular phosphate export, which resulted in increased intracellular ATP levels. The in vivo inhibition of IP6K decreased plasma phosphate levels without inhibiting gut intake or kidney reuptake of phosphate, demonstrating a pivotal role of IP6K-regulated cellular phosphate export on circulating phosphate levels. IP6K inhibition-induced decrease in intracellular inositol pyrophosphate, an enzymatic product of IP6K, was correlated with phosphate changes. Chronic IP6K inhibition alleviated hyperphosphataemia, increased kidney ATP, and improved kidney functions in chronic kidney disease rats. Our results demonstrate that the enzymatic activity of IP6K regulates circulating phosphate and intracellular ATP and suggest that IP6K inhibition is a potential novel treatment strategy against hyperphosphataemia.

[1] Research Division, SCOHIA PHARMA Inc, Kanagawa, Japan. ✉email: yusuke.moritoh@scohia.com

nositol hexakisphosphate kinases (IP6K1, 2, and 3 in mammals) participate in diverse cellular signalling pathways by generating inositol pyrophosphates from their precursors[1,2]. Inositol pyrophosphates are highly energetic molecules, and the additional phosphorylation of their phosphorylated inositol hydroxyl group is catalysed by two classes of enzymes: IP6Ks with 5-kinase activity and diphosphoinositol pentakisphosphate kinases (PPIP5Ks) with 1-kinase activity generates $5\text{-InsP}_7$ and its downstream molecule $\text{InsP}_8$, respectively, neither of which is a fully characterized inositol pyrophosphate[1,2]. PPIP5Ks generate $1\text{-InsP}_7$ from $\text{InsP}_6$; however, this form of $\text{InsP}_7$ makes minor contributions to the total $\text{InsP}_7$ levels in mammalian cells[3–6], which suggests that IP6K-mediated $5\text{-InsP}_7$ generation is the predominant $\text{InsP}_7$ pathway leading to $\text{InsP}_8$ (Fig. 1a). Recent experimental observations suggest that inositol pyrophosphates are the conserved regulators for phosphate homeostasis across species[7]. The levels of inositol pyrophosphates are correlated with phosphate availability in yeast, fungi, plants, and animals[8,9]. Several proteins involved in maintaining phosphate homeostasis contain SPX (Syg1/Pho81/*XPR1*) domain[10,11]. The binding of inositol pyrophosphates to the SPX domain of target protein regulates phosphate homeostasis[9], which indicates that inositol pyrophosphates and protein-equipping SPX domain likely act together to control phosphate levels.

IP6K1 and IP6K2 have been shown to regulate phosphate homeostasis in cells[12], and $\text{InsP}_7$ may respond to shifts in extracellular phosphate status in vitro[13]. The single-nucleotide polymorphisms in IP6K3 gene locus are associated with differences in serum phosphate concentration in humans[14,15]. Additionally, Li et al. have shown that $\text{InsP}_8$, which functions further downstream of $\text{InsP}_7$, regulates phosphate levels in cells in vitro[16]. However, the physiological roles of IP6K and inositol pyrophosphate in regulating phosphate homeostasis in vivo remain unknown in mammals. In addition, *Ip6k1* knockout (KO) cells show increased intracellular ATP levels[17]; however, the mechanism underlying this effect and its in vivo relevance are not clear.

Therefore, herein, we aimed to investigate the physiological importance and molecular mechanisms underlying the regulation of phosphate and intracellular ATP levels by IP6K in mammals in vivo. We believe that the findings of this study could facilitate the development of new therapeutic strategies.

## Results

**SC-919: potent, selective IP6K inhibitor.** IP6K mediates the synthesis of $5\text{-InsP}_7$ from $\text{InsP}_6$, and potentially $\text{InsP}_8$ from $1\text{-InsP}_7$. $5\text{-InsP}_7$ is a predominant $\text{InsP}_7$ in mammals (Fig. 1a). We

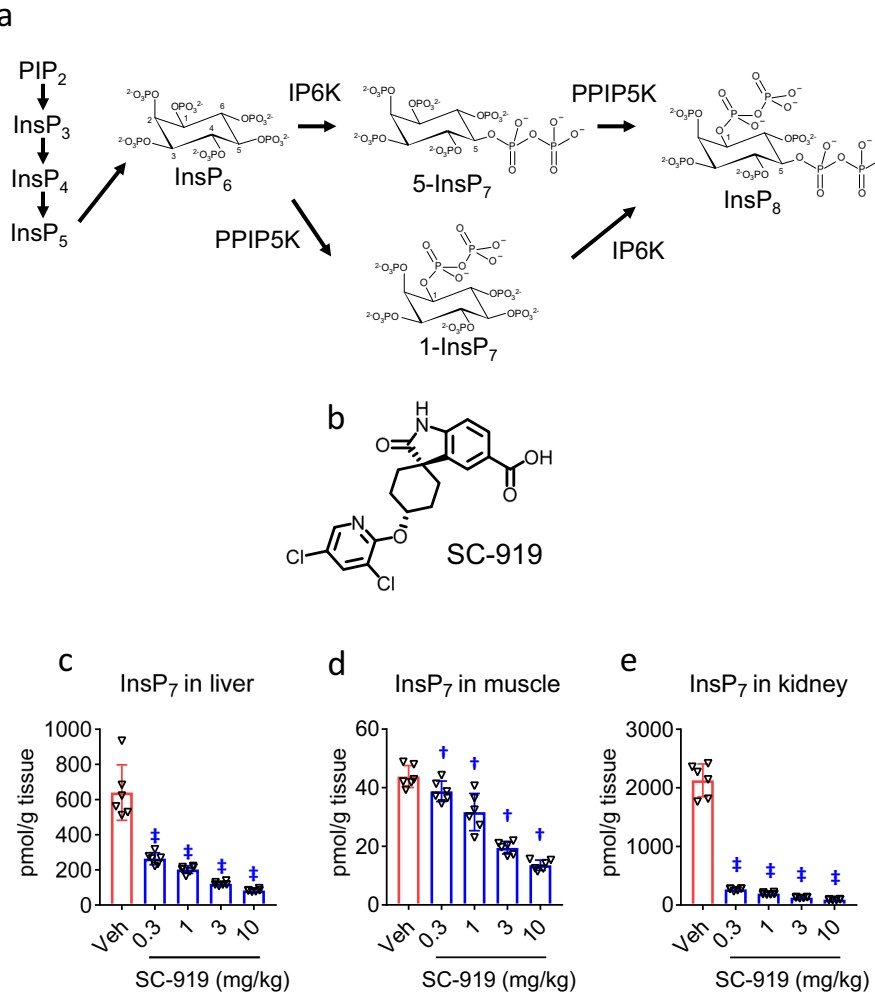

**Fig. 1 SC-919 is an effective inhibitor of IP6K in vivo. a** IP6K-mediated synthesis of the inositol pyrophosphates. $\text{PIP}_2$ phosphatidylinositol 4,5-bisphosphate, Ins inositol; the subscripts denote the total number of phosphates; PPIP5K diphosphoinositol pentakisphosphate kinase. **b** The chemical structure of SC-919. $\text{InsP}_7$ levels in the liver (**c**), muscle (**d**), and kidney (**e**) after 2 h of SC-919 administration to rats. Values indicate mean ± S.D. ($n = 6$ biological replicates). [†]$P < 0.05$ and [‡]$P < 0.05$ vs. vehicle as determined using Williams' test and Shirley–Williams test, respectively. Veh vehicle.

**Table 1 Inhibitory activity of SC-919 on human IP6K1, human IP6K2, and human IP6K3 kinase activity.**

| Test material | IC$_{50}$ (nmol/L) | | |
|---|---|---|---|
| | IP6K1 | IP6K2 | IP6K3 |
| SC-919 | <5.2[a] | <3.8[a] | 0.65 ± 0.13 |
| TNP | 270 ± 100 | 850 ± 330 | 260 ± 21 |

The ADP-glo assay condition was performed as follow: The compounds were co-incubated with recombinant human IP6K1 (15 nM), human IP6K2 (12 nM), or human IP6K3 (1.5 nM) in the assay buffer. Following the addition of ATP and InsP$_6$ at final concentrations of 15 and 50 μM, respectively, a kinase reaction was carried out for 2 h. Thereafter, the ADP-Glo™ kinase assay (Promega, Madison, WI, USA) was performed.
[a]SC-919 showed inhibitory activity reaching detection limits for IP6K1 assay and IP6K2 assay, respectively. Values indicate mean ± S.D. (n = 3 biological replicates of 4–8 technical replicates).

**Table 2 In vitro plasma protein binding ratios of SC-919 in rats, monkeys, and humans.**

| Matrix | Concentration of compound (μg/mL) | Binding ratio (%) |
|---|---|---|
| Rat | 5 | 99.9 |
| | 0.5 | 99.9 |
| | 0.05 | >99.6 |
| Monkey | 5 | 99.0 |
| | 0.5 | 99.1 |
| | 0.05 | 99.2 |
| Human | 5 | 99.8 |
| | 0.5 | 99.7 |
| | 0.05 | >99.6 |

The data are shown as the mean of three samples. Matrices shown are pooled plasma samples of 19 male SD rats (8 weeks old), pooled plasma samples of 10 male monkeys (2 years 7 months to 3 years 4 months), and pooled plasma samples of 5 male and 5 female humans.

performed high-throughput screening and compound optimization to identify a potential IP6K inhibitor that suppresses inositol pyrophosphate production to investigate the physiological functions of IP6K in vitro and in vivo[18]. As a result, we identified SC-919 (Fig. 1b and Supplementary Fig. 1)[18], which potently inhibited IP6K1, 2, and 3 (Table 1), and this inhibitory activity was much more potent than that of *N2*-(m-(trifluoromethyl)benzyl) *N6*-(p-nitrobenzyl)purine (TNP), a commercially available IP6K inhibitor[4] (Table 1). Further, we examined the kinome selectivity profile of 1 μM SC-919 to assess its selectivity window using a scanEDGE Kinase Assay Panel. SC-919 did not inhibit other kinases, indicating its selectivity for IP6K (Supplementary Table 1). The protein binding study revealed that the plasma protein binding ability of SC-919 was 99.0–99.9% in rats, monkeys, and humans (Table 2). To determine whether SC-919 reduced intracellular InsP$_7$ levels, we developed a novel liquid chromatography-tandem mass spectrometry (LC-MS/MS)-based InsP$_7$ (total InsP$_7$; 5-InsP7 + 1-InsP$_7$) measurement method to facilitate the quantification of InsP$_7$ levels in vitro and in vivo without radioisotope labelling (Supplementary Fig. 2 (analytical results of InsP$_6$.$d_6$ and InsP$_7$), methodological qualification results are shown in Supplementary Table 2 (calibration curve), Supplementary Table 3 and Supplementary Table 4 (accuracy and precision), and Supplementary Fig. 3 and Supplementary Fig. 4 (representative chromatogram)). Oral administration of SC-919 resulted in a dose-dependent decrease in InsP$_7$ levels in the liver, skeletal muscle, and kidney of normal rats (Fig. 1c–e, tissue SC-919 levels are included in Supplementary Table 5), and InsP$_6$ levels exhibited minor changes (Supplementary Fig. 5). SC-919 is an orally available selective IP6K inhibitor that is highly effective in vivo.

**IP6K regulates phosphate export via *XPR1*.** SC-919 administration to rats reduced plasma phosphate levels. We hypothesized that SC-919 regulates the transmembrane movement of phosphate. To test this, we treated HEK293 cells (called 293 cells henceforth) with SC-919 for 4 h and evaluated the export and uptake of phosphate using $^{32}$P-labelled phosphate ($^{32}$P-phosphate). SC-919 decreased the levels of InsP$_7$ in 293 cells (Fig. 2a; InsP$_6$ showed only a minor change, Supplementary Fig. 5). Further, the export of $^{32}$P activity from the intracellular to extracellular spaces was suppressed by SC-919, which was associated with the reduction in InsP$_7$ levels with a marginal change in intracellular $^{32}$P activity in phosphate export assay (Fig. 2b, c). This finding was similar with that of a previous study using TNP[19]. Additionally, phosphate uptake assays revealed elevated intracellular $^{32}$P activity (Fig. 2d), which was different from that observed in a previous study using TNP[19]. These observations indicate that SC-919-mediated IP6K inhibition suppressed phosphate export and/or stimulated phosphate uptake. Inositol

pyrophosphates regulate several proteins involved in phosphate homeostasis by binding to the SPX domain of target proteins across species[7]. The SPX-domain-containing *XPR1* protein mediates the export of phosphate in metazoans[20]. *IP6K1/K2* double KO cells exhibit reduced phosphate uptake and export[12]. To determine whether phosphate uptake or export is controlled by pharmacological inhibition of IP6K, we evaluated the role of *XPR1*, a phosphate exporter, in the effects induced by IP6K inhibition in cells. SC-919 decreased the levels of InsP$_7$ in wild-type and *XPR1* KO cells (Fig. 2e, f; InsP$_6$, Supplementary Fig. 5; cell profiles for *XPR1* KO cells, Supplementary Fig. 6). SC-919 decreased the export of $^{32}$P activity in wild-type cells with a marginal change of intracellular $^{32}$P activity in phosphate export assay; however, this effect was abolished in *XPR1* KO cells (Fig. 2g, h). Additionally, phosphate uptake assays showed that SC-919 increased intracellular levels of $^{32}$P activity in wild-type cells, and this effect was abrogated in *XPR1* KO cells (Fig. 2i), indicating that the inhibition of *XPR1*-mediated phosphate export, rather than stimulated phosphate uptake, increased $^{32}$P activity in cells. To determine whether the effects induced by IP6K inhibition are mediated via *XPR1*, a gain-of-function experiment was conducted in *XPR1* KO cells. The introduction of wild-type and SPX-deleted *XPR1*, which lacks inositol pyrophosphate binding region, a SPX domain, of this transporter with unmodified transmembrane regions, increased the export of $^{32}$P activity and decreased intracellular $^{32}$P activity (Fig. 2j, k), indicating that both the *XPR1* forms export phosphate into extracellular spaces. The IP6K-inhibition-mediated suppression of $^{32}$P activity export was rescued by the introduction of wild-type *XPR1*, but not by SPX-deleted *XPR1* in *XPR1* KO cells (Fig. 2j), demonstrating that the SPX domain of *XPR1* is critical for inducing these effects. Moreover, SC-919 did not influence protein levels in cells for up to 4 h of treatment in vitro (Supplementary Table 6).

**IP6K inhibition reduces circulating phosphate levels.** Following the oral administration to normal rats, plasma SC-919 concentrations increased in a dose-dependent manner (Fig. 3a). SC-919 administration decreased plasma phosphate levels in normal rats (Fig. 3b) and was correlated with reduced InsP$_7$ in tissues (Fig. 1c–e). To determine whether this effect is preserved in non-human primate, we tested the oral administration of SC-919 in normal monkeys, and showed that 0.3 and 3 mg/kg SC-919 was significantly effective in lowering the plasma phosphate levels (Fig. 3c–e). Thus, IP6K physiologically regulates phosphate levels in rodents and monkeys. In addition to the cellular regulation of phosphate, plasma phosphate is regulated by uptake in the gut

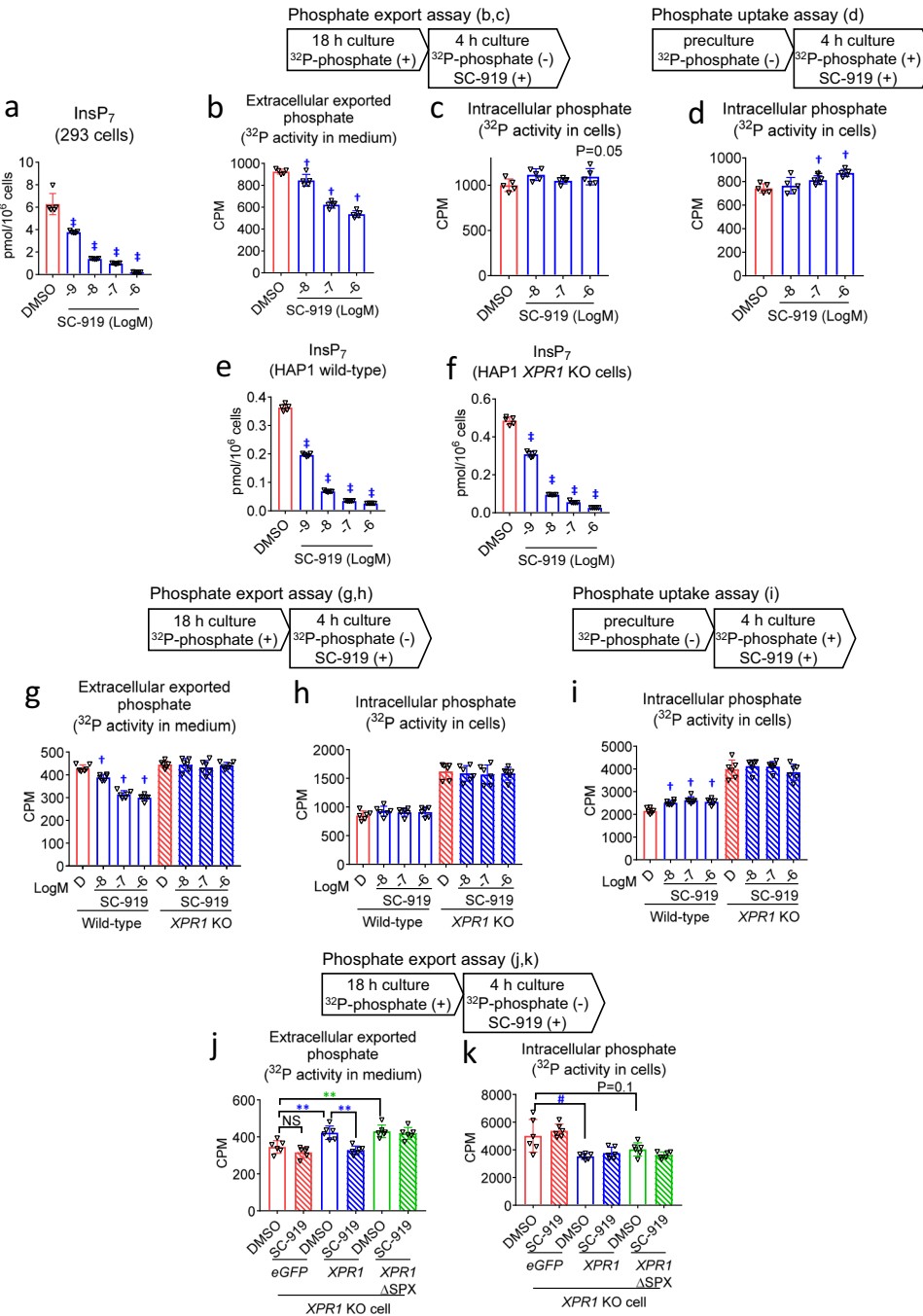

**Fig. 2 IP6K inhibition reduces the export of phosphate via an *XPR1*-dependent mechanism involving the SPX domain.** Effect of treatment with SC-919 for 4 h on the **a** intracellular InsP$_7$ levels in 293 cells, **b** exported extracellular $^{32}$P activity, and **c** intracellular $^{32}$P activity in the phosphate export assay in 293 cells, **d** intracellular $^{32}$P activity in the phosphate uptake assay in 293 cells, **e, f** Levels of InsP$_7$ in HAP1 wild-type and *XPR1* KO cells, **g** levels of exported extracellular $^{32}$P activity, and **h** intracellular $^{32}$P activity in phosphate export assay in HAP1 wild-type and *XPR1* KO cells, **i** intracellular $^{32}$P activity in phosphate uptake assay in HAP1 wild-type and *XPR1* KO cells, **j, k** Exported extracellular $^{32}$P activity and intracellular $^{32}$P activity in phosphate export assay in *XPR1* KO cells introduced with either *eGFP*, *XPR1*, or inositol pyrophosphates-binding site-deleted *XPR1* (*XPR1*ΔSPX) treated with either DMSO or SC-919 (1 μM). Values indicate mean ± S.D. (*n* = 5 biological replicates for **a**–**f** and 6 biological replicates for **g**–**k**). †*P* < 0.05 and ‡*P* < 0.05 vs. vehicle as determined using Williams' test and Shirley–Williams test, respectively. \*\**P* < 0.01 as determined using Student's *t*-test, and #*P* < 0.05 as determined using the Aspin–Welch test. D, DMSO. CPM, counts per minute.

and excretion through urine[21]. To explore the potential effects of IP6K inhibition on these parameters, $^{32}$P-phosphate was administered orally to SC-919-treated rats, and the effects on phosphate excretion via faeces and urine were tested after the first and seventh doses of SC-919. The $^{32}$P activity in faeces was slightly higher in SC-919-treated rats than in vehicle-treated rats;

however, the change was not significant (Fig. 3f). Lanthanum carbonate[21], a phosphate binder, increased phosphate excretion in faeces (Fig. 3f). The excretion of $^{32}$P activity via urine was lower in SC-919-treated rats than in vehicle-treated rats (Fig. 3g), which is most likely due to the decreased plasma phosphate levels.

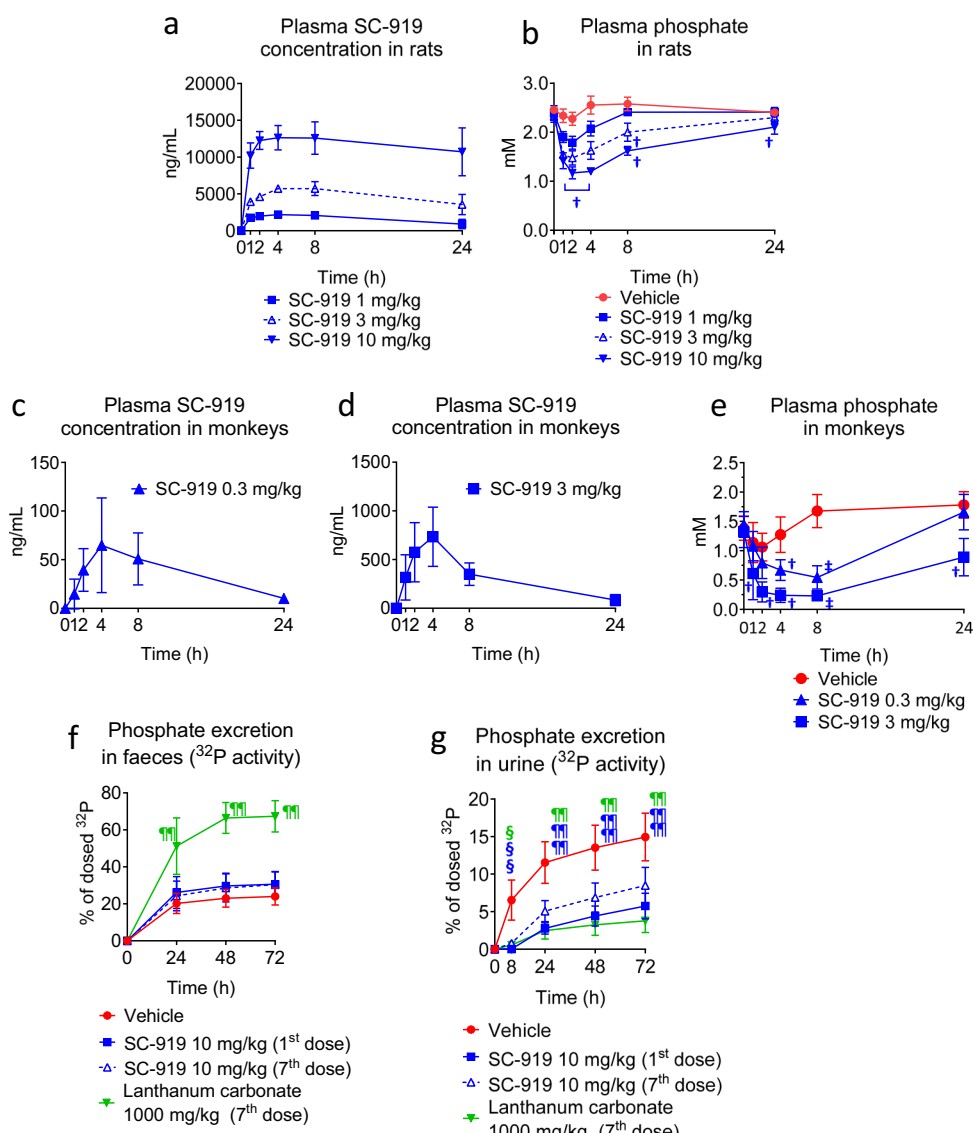

**Fig. 3 IP6K inhibition decreases plasma phosphate levels in rats and monkeys and does not inhibit phosphate intake by the gut or phosphate reuptake by the kidney in rats. a** Plasma concentration of SC-919 and **b** phosphate levels in normal rats administered SC-919. ($n = 3$ biological replicates for pharmacokinetics and $n = 5$ biological replicates for the determination of plasma phosphate levels). **c, d** Concentration of SC-919 and **e** phosphate levels in the plasma of normal monkeys administered SC-919. ($n = 6$ biological replicates). **f** Effect of SC-919 on the cumulative $^{32}P$ activity in the faeces of rats that were administered $NaH_2^{32}PO_4$ solution. ($n = 5$ biological replicates). **g** Effect of SC-919 on the cumulative $^{32}P$ activity in the urine of rats that were administered $NaH_2^{32}PO_4$ solution. ($n = 5$ biological replicates). Values indicate mean ± S.D. $^{\dagger}P < 0.05$ and $^{\ddagger}P < 0.05$ vs. vehicle as determined using Williams' test and Shirley–Williams test, respectively. $^{\P\P}P < 0.01$ and $^{\S}P < 0.05$ vs. vehicle as determined using Dunnett's test and Steel test, respectively.

**IP6K inhibition alleviates hyperphosphataemia.** Based on the results using normal animals, we tested the efficacy of SC-919 in rodent models of hyperphosphataemia. Adenine-treated rats are a model for chronic kidney disease in humans, as the resulting decrease in renal excretion of phosphate induces hyperphosphataemia[22]. A single oral dose of SC-919 effectively decreased the plasma phosphate levels of adenine-treated rats with hyperphosphataemia (Fig. 4a, b). Sub-chronic dosing of SC-919 for 7 days indicated that the inhibition of IP6K showed sustained efficacy in reducing plasma phosphate levels in adenine-treated rats (Fig. 4c, d). As hyperphosphataemia is evident during the late stages of kidney function impairment[21], we explored the therapeutic efficacy of SC-919 in rats with severely impaired kidneys. The oral administration of SC-919 decreased the phosphate levels in the plasma of bilaterally nephrectomized rats (Fig. 4e, f).

**IP6K inhibition increases ATP levels.** IP6K inactivation causes an increase in ATP levels, and there are several hypotheses for the mechanisms underlying this effect[2,17]. The treatment of 293 cells with SC-919 increased the intracellular ATP levels (Fig. 5a), and we examined the role of *XPR1* in mediating this effect. Basal intracellular ATP levels were higher in *XPR1* KO cells than in wild-type cells (Fig. 5b), and the SC-919-induced increase in ATP levels was abolished in *XPR1* KO cells (Fig. 5c), suggesting that this effect is mediated by *XPR1*. The introduction of wild-type *XPR1* in *XPR1* KO cells resulted in a decrease in the basal levels of ATP (Fig. 5d). Additionally, the introduction of SPX-deleted *XPR1*, which could export phosphate (Fig. 2j), decreased the basal intracellular levels of ATP (Fig. 5d). These results suggest that *XPR1*-mediated regulation of phosphate export determines intracellular ATP levels. Moreover, although SC-919 was associated with elevated ATP levels in wild-type *XPR1*-introduced

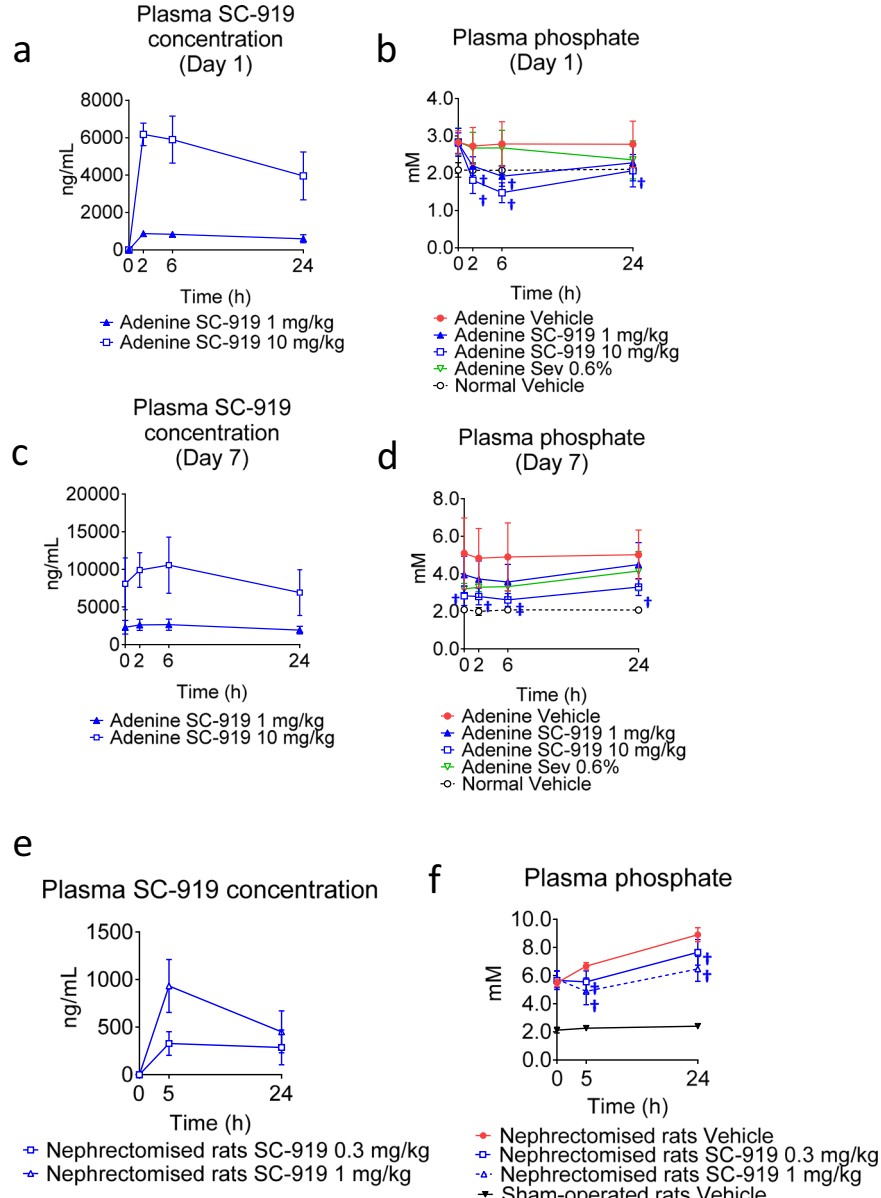

**Fig. 4 IP6K inhibition decreases the plasma phosphate levels in rats with hyperphosphataemia. a** Concentration of SC-919 and **b** plasma phosphate levels following the initial dose of SC-919 or sevelamer in adenine-treated rats. ($n = 3$ biological replicates for pharmacokinetics and $n = 6$ and 5 biological replicates for adenine-treated rats and normal rats for determination of phosphate levels in the plasma). **c** Concentration of SC-919 and **d** plasma phosphate levels following the seventh dose of SC-919 or sevelamer in adenine-treated rats. ($n = 3$ biological replicates for pharmacokinetics and $n = 6$ and 5 biological replicates for adenine-treated rats and normal rats for determination of phosphate levels in the plasma). **e** Concentration of SC-919 and **f** plasma phosphate levels in bilaterally nephrectomized rats that were orally administered with SC-919. Animals found dead during the study (1 rat in SC-919 1 mg/kg of **e**, 2 rats in nephrectomized-vehicle, and 1 rat in nephrectomized-SC-919 1 mg/kg of **f**. ($n = 5-7$ biological replicates for nephrectomzed rats, and $n = 6$ biological replicates for sham-operated rats for **e** and **f**). Values indicate mean ± S.D. $^{\dagger}P < 0.05$ and $^{\ddagger}P < 0.05$ vs. vehicle as determined using a Williams' test and the Shirley–Williams test, respectively. Sev sevelamer.

*XPR1* KO cells, it failed to induce a similar change in SPX-deleted *XPR1*-introduced *XPR1* KO cells (Fig. 5d), demonstrating that the SPX domain of *XPR1*, an inositol pyrophosphate binding region, is essential for inducing IP6K-inhibition-mediated ATP elevation. To obtain direct evidence that IP6K-inhibition-mediated phosphate regulation is critical to the increase in intracellular ATP levels, we traced $^{32}$P-labelled ATP in cells treated with $^{32}$P-phosphate and SC-919. To this end, we isolated the ATP fraction from cells treated with $^{32}$P-phosphate and SC-919 via HPLC and determined the $^{32}$P activity. SC-919 caused an increase in the $^{32}$P activity in the ATP fraction in cells pre-treated with $^{32}$P-phosphate for 18 h, which supports the role of intracellular $^{32}$P-

phosphate in the increase in cellular ATP levels (Fig. 5e). Further, SC-919 increased the $^{32}$P activity in the ATP fraction in cells treated with $^{32}$P-phosphate during the 4 h exposure to SC-919, indicating a role of $^{32}$P-phosphate uptake in the increase of intracellular ATP levels (Fig. 5f).

**IP6K inhibition is therapeutically effective in chronic kidney disease (CKD).** To evaluate the chronic therapeutic effects of IP6K inhibition, SC-919 was administered to adenine-treated rats for 5 weeks (Experimental protocol is shown in Fig. 6a). The effects of IP6K were compared with those induced by sevelamer as a positive control, which binds to phosphate in the gut to

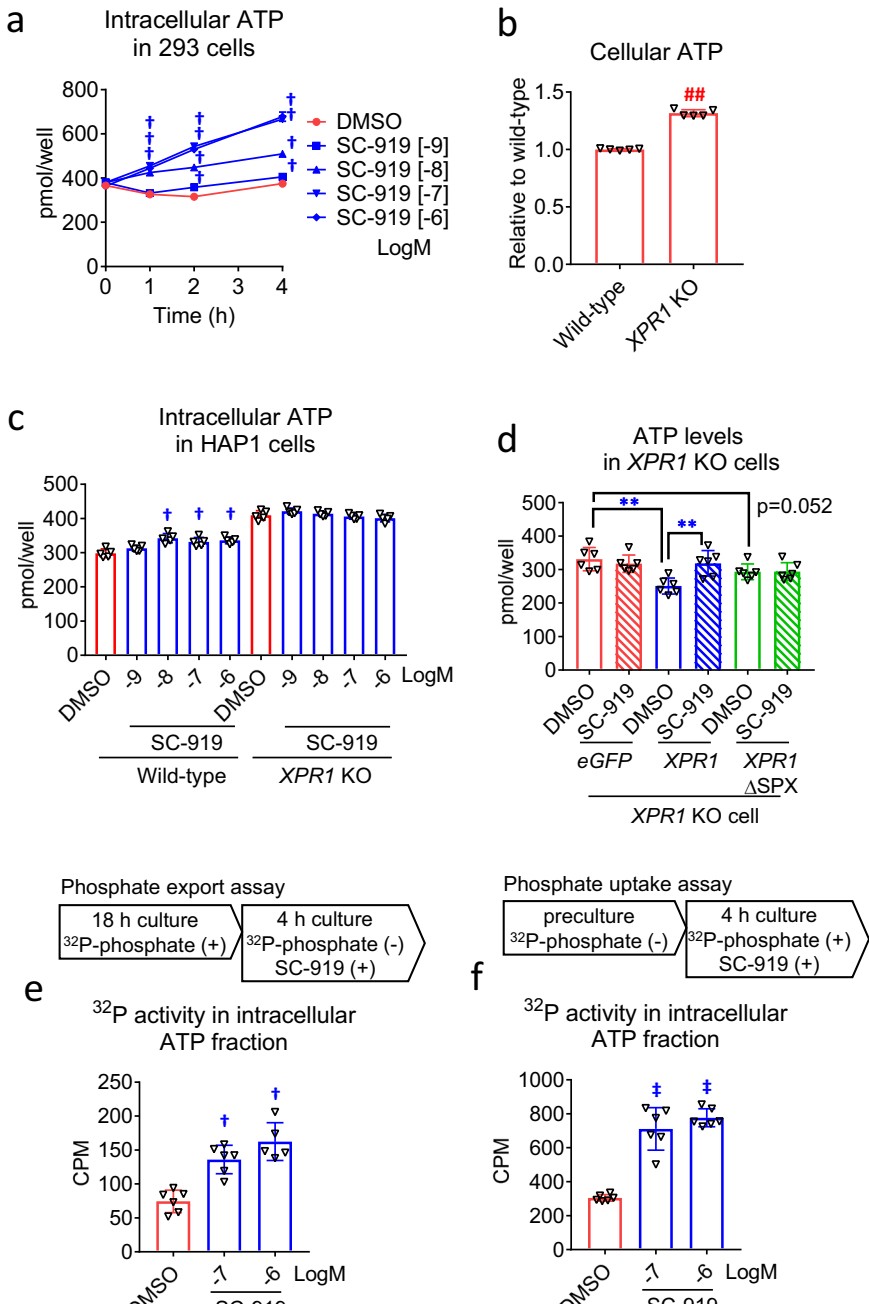

**Fig. 5 IP6K inhibition increases the intracellular levels of ATP in an *XPR1*-dependent manner. a** Time course of ATP levels in SC-919-treated 293 cells. **b** Intracellular levels of ATP in HAP1 wild-type and *XPR1* KO cells. Effects of the treatment with SC-919 after 4 h on **c** intracellular levels of ATP in HAP1 wild-type and *XPR1* KO cells, **d** intracellular levels of ATP in *XPR1* KO cells introduced with either *eGFP*, *XPR1*, or inositol pyrophosphates-binding site-deleted *XPR1* (*XPR1*ΔSPX) treated with either DMSO or SC-919 (1 μM), **e** $^{32}$P activity in ATP in 293 cells pre-treated with $^{32}$P-phosphate followed by SC-919 treatment, **f** $^{32}$P activity in ATP in 293 cells pre-treated with non-labelled phosphate followed by SC-919 plus $^{32}$P-phosphate treatment. Values indicate mean ± S.D. ($n = 5$ biological replicates for **a–c**, 5–6 biological replicates for **e**, and 6 biological replicates for **d, f**). $^{†}P < 0.05$ and $^{‡}P < 0.05$ vs. vehicle as determined using Williams' test and Shirley–Williams test, respectively. $^{**}P < 0.01$ as determined using Student's $t$-test, and $^{##}P < 0.01$ as determined using the Aspin–Welch test. CPM, counts per minute.

consequently decrease phosphate levels in the plasma[21]. No abnormal responses or gastrointestinal symptoms were observed in rats administered SC-919. Treatment with SC-919 improved food intake behaviour and increased the body weight of the rats (Fig. 6b, c), which are impaired in this model. The adenine-treated rats exhibited high levels of GDF15 (Fig. 6d), which is a stress response cytokine and induces reduced food intake and weight loss[23]. Consistent with the increase in food intake, levels of

circulating GDF15 significantly decreased in rats administered SC-919 (Fig. 6d). Abnormal hormone profiles, which are similar to those in patients with CKD, were observed in the adenine-treated rats (Fig. 6e–g). Chronic dosing with SC-919 decreased the levels of plasma FGF23 (Fig. 6e), an endocrine hormone that stimulates the excretion of phosphate by the kidneys and is increased in the presence of CKD[24]. The administration of SC-919 also increased 1,25(OH)$_2$-D, which increases the absorption

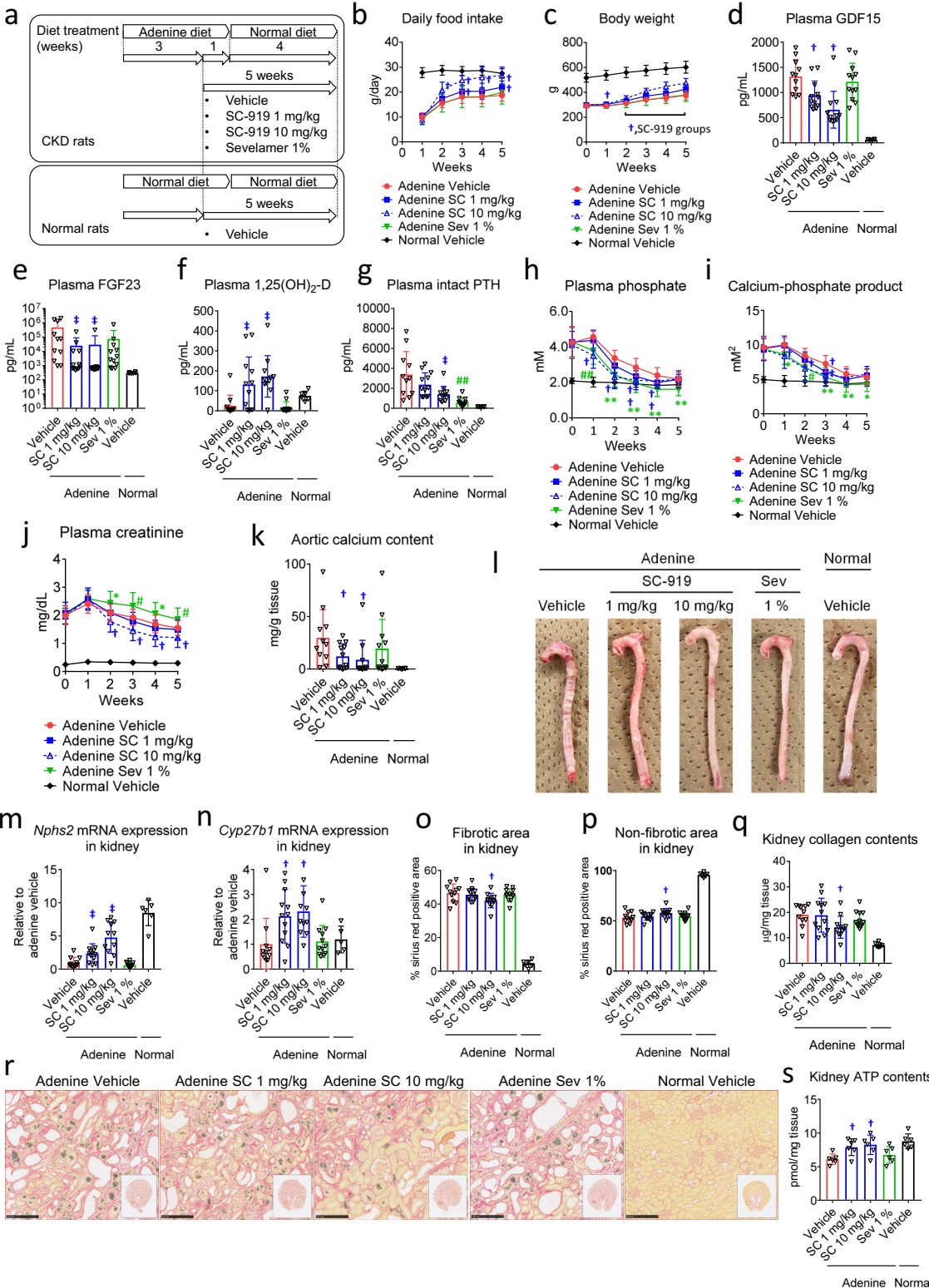

**Fig. 6 Chronic IP6K inhibition alleviates CKD-induced hyperphosphataemia and related parameters in adenine-treated rats.** Evaluation of the chronic effect of SC-919 in adenine-treated rats. **a** Experimental protocol for **a–r**. **b** Daily food intake. **c** Body weight. **d** Plasma GDF15 levels. **e** Plasma FGF23 levels. **f** Plasma 1,25(OH)$_2$-D levels. **g** Plasma intact PTH levels. **h** Plasma phosphate levels. **i** Plasma calcium-phosphate products. **j** Plasma creatinine levels. **k** Aortic calcium content. **l** Representative aortic images. Levels of the **m** *Nphs2* and **n** *Cyp27b1* mRNAs in the kidney. **o** Fibrotic and **p** non-fibrotic area in kidney. **q** Kidney collagen contents. **r** Representative kidney images (bar = 250 μm, each lower right square shows the entire kidney image). Animals found dead during the study (1 rat in vehicle, 1 rat in SC-919 10 mg/kg). (*n* = 11–12 biological replicates for adenine-treated rats and *n* = 5–6 biological replicates for normal rats for **a–r**). **s** Kidney ATP levels (6 h after the drug dose) in adenine-rats treated with SC-919 for 8 days. (*n* = 6 biological replicates). Values indicate mean ± S.D. †*P* < 0.05 and ‡*P* < 0.05 vs. vehicle as determined using Williams' test and Shirley–Williams test, respectively. *\*P* < 0.05 and *\*\*P* < 0.01 vs. vehicle as determined using Student's *t*-test and #*P* < 0.05 and ##*P* < 0.01 vs. vehicle as determined using the Aspin–Welch test. Sev, sevelamer.

of calcium and phosphate in the gut (Fig. 6f), which is decreased in CKD[24], and decreased levels of parathyroid hormone, which regulates the calcium and phosphate levels in the serum (Fig. 6g), which are increased in CKD[24]. SC-919 reduced plasma phosphate levels (Fig. 6h) and calcium-phosphate product (Fig. 6i), which is a clinically relevant tool for estimating the cardiovascular risk of patients with renal failure[25]. SC-919 decreased plasma creatinine levels (Fig. 6j), which is a well-established marker for renal dysfunction[26], while vascular calcification was significantly alleviated by SC-919 (Fig. 6k, l). Furthermore, SC-919 increased the levels of the *Nphs2* and *Cyp27b1* mRNAs (Fig. 6m, n), both of which are essential for glomerular function (Uniprot, Q8K4G9) and the generation of the active form of vitamin D (Uniprot, O35132), suggesting an improvement in kidney function. Imaging analysis showed that SC-919 decreased fibrotic area, increased non-fibrotic area, and reduced collagen content in kidney (Fig. 6o–r). Finally, SC-919 rescued the ATP levels in the kidneys of adenine-treated rats to almost normal levels after 8 days of treatment (Fig. 6s). Although, sevelamer decreased phosphate levels, the observations related to food intake, body weight, and the levels of GDF15, FGF23, $1,25(OH)_2$-D, plasma creatinine, aortic calcium content, and the mRNAs for *Nphs2* and *Cyp27b1*, kidney fibrosis, and kidney ATP levels were specific to SC-919 treatment.

## Discussion

This study illustrates the physiological significance of phosphate regulation by IP6K in vivo and provides a novel strategy to treat hyperphosphataemia and the associated complications using IP6K inhibitor-mediated suppression of cellular phosphate export (Fig. 7).

Gene deletion of each of the three IP6K subtypes, IP6K1, IP6K2, and IP6K3, results in unique phenotypes in vivo[1]; however, no KO mouse studies have yet demonstrated the role of IP6K activity on circulating phosphate levels in vivo. This may be a result of the multiple mechanisms of the body operating over time to maintain blood phosphate levels within a narrow range in vivo[27]. Previous genetic studies have shown that *IP6K1/2* KO cells exhibit reduced phosphate uptake and export in vitro[12]. In our study, selective IP6K inhibition by SC-919 decreased cellular $InsP_7$ levels and phosphate export; however, SC-919-mediated IP6K inhibition did not inhibit phosphate uptake in cells. Exactly how phosphate export from cells regulates plasma phosphate levels remain to be fully characterized. Using a highly selective inhibitor against IP6K, which could be administered and examined in vivo, the present study demonstrates the contribution of IP6K-mediated cellular phosphate export on circulating phosphate concentrations in mammals. Considering that the fluctuation of cellular inositol pyrophosphate induces cellular events within minutes to hours, the use of an in vivo-compatible small-molecule inhibitor could help illustrate the physiological role of IP6K-generated inositol pyrophosphate on phosphate regulation in vivo.

The present study further suggests that the inhibition of phosphate uptake by the gut is not a primary contributor to the IP6K-inhibition-induced reduction of circulating phosphate levels. Additionally, the inhibition of phosphate reuptake in the kidney was not induced by IP6K inhibition. Therefore, inhibition of the *XPR1*-mediated export of cellular phosphate may be the primary mechanism through which IP6K inhibition decreases plasma phosphate levels in vivo.

Conditional genetic deletion of *XPR1* in the renal tubule resulted in phosphaturia[28], which is in contrast with the IP6K inhibition decreasing plasma phosphate levels and phosphate excretion in the urine. Considering that plasma phosphate levels

directly influence urine phosphate excretion[29], the exact role of IP6K-inhibition-regulated *XPR1* in the kidney should be studied in mice in which normal plasma phosphate levels are maintained through phosphate infusion.

We have demonstrated that the presence of the SPX domain in *XPR1* is indispensable for the inhibition of IP6K to prevent phosphate export, which is consistent with the results of recent studies[12,19]. Previous studies have shown that $InsP_7$, generated by IP6K and $InsP_8$ (a molecule downstream of $InsP_7$ generated by PPIP5K), binds to the SPX domain of human *XPR1*[12,16]. Notably, both $InsP_7$ and $InsP_8$ may be responsive to extracellular phosphate concentrations in cells[13]. The lack of IP6K activity in inositol pyrophosphate biosynthesis pathways results in the loss of both $InsP_7$ and $InsP_8$. Therefore, $InsP_7$ and/or $InsP_8$ may contribute to the regulation of *XPR1* via binding to its SPX domain in vivo. More recently, Li et al. have demonstrated that $InsP_8$ plays a major role in regulating cellular phosphate levels in cells[16]. Therefore, the reduction in circulating phosphate levels by IP6K inhibition may be mediated by a decrease in $InsP_8$ via $InsP_7$ reduction in vivo. Additional studies on the role of inositol pyrophosphate in regulating phosphate levels in vivo are required to validate this hypothesis.

The inhibition of IP6K using SC-919 to suppress cellular phosphate export has therapeutic relevance, as we found that IP6K inhibition alleviated hyperphosphataemia and improved kidney function parameters. Suppressing cellular phosphate export owing to IP6K inhibition also resulted in increased ATP levels, which fully depended on the SPX domain of *XPR1*, suggesting a critical role of inositol pyrophosphates downstream to IP6K. ATP is a key small molecule required for various cellular signalling pathways/events, and its levels are decreased in the kidneys of patients with CKD[30]. Therefore, an investigation into the effect of an increase in ATP levels on homeostasis and kidney markers in the presence of CKD is clinically relevant. Chronic treatment with SC-919 decreased GDF15, which is induced in response to various metabolic stress factors[31], and caused an increase in the food intake, which decreased in adenine-treated rats. The SC-919-induced increase in tissue ATP levels may reduce tissue stress, thereby decreasing the levels of GDF15; however, this remains to be confirmed. As a result of the increased intake of food, the body weight of rats increased following SC-919 treatment. As wasting/cachexia is prevalent among patients with CKD[32], suggesting that the effects of SC-919 administration would be therapeutically beneficial for these patients. Furthermore, inhibiting IP6K improved kidney functions in CKD, as shown by the decreased creatinine levels in plasma and presence of improved kidney function markers, which were not observed with treatment of the phosphate binder. This may also have contributed to the alleviation of hyperphosphataemia. Together, these results suggest that SC-919 has significant potential to clinically treat hyperphosphataemia and improve kidney function in patients with CKD.

The non-labelled, absolute, and highly sensitive measurement of $InsP_6$ and $InsP_7$ is critical to confirm the pharmacological effect of SC-919 on IP6K inhibition both in vitro and in vivo. Compared to previous methods[33,34], our LC-MS/MS method detected higher levels of $InsP_6$ and $InsP_7$ in cells and tissues, suggesting a better sensitivity. Among tissues tested in this study, the kidney showed the highest levels of $InsP_7$, and comparable $InsP_6$ levels with the liver. In contrast, skeletal muscle showed lower $InsP_7$ and $InsP_6$ levels. To date, absolute levels of $InsP_6$ and $InsP_7$ in each organ in vivo were largely unknown due to the methodological limitation. Thus, determining the $InsP_6$ and $InsP_7$ levels and how they are regulated in each organ remain to be investigated.

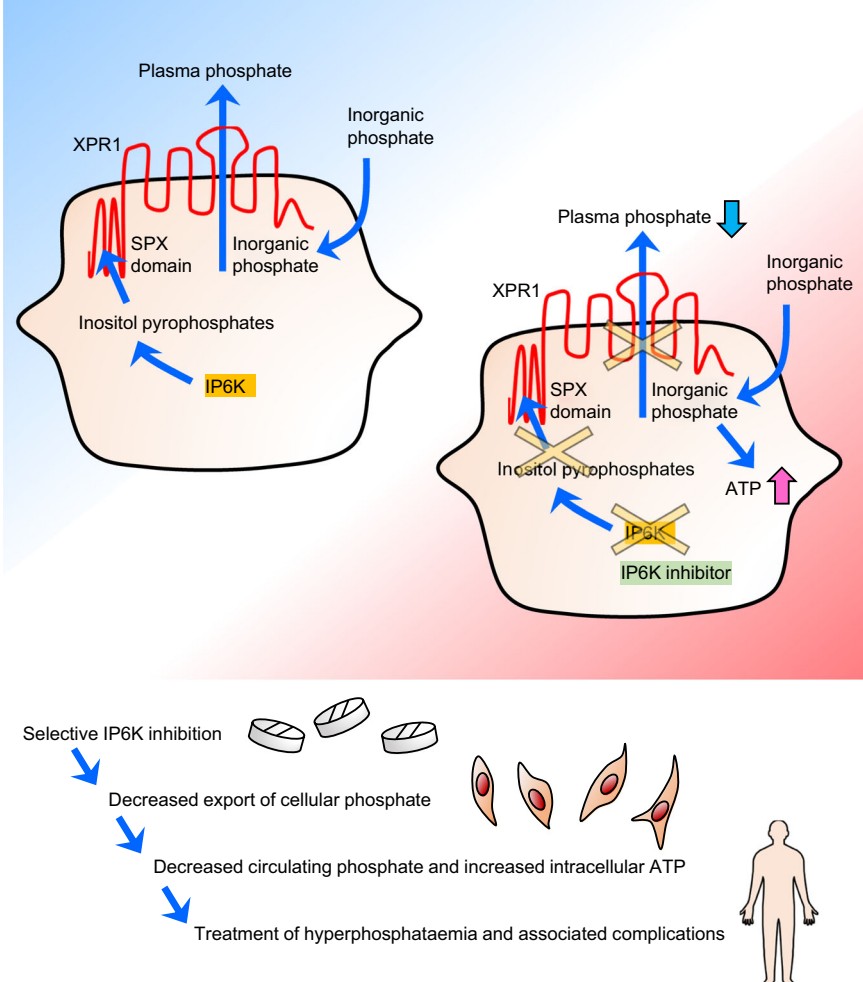

**Fig. 7 Schematic representation of IP6K-mediated phosphate regulation and its therapeutic relevance.** In the normal condition, IP6K stimulates cellular phosphate export via SPX domain of *XPR1*, maintaining plasma phosphate levels in vivo. IP6K-inhibition-mediated reduction of InsP$_7$ inhibits phosphate export ability of *XPR1* in cells, resulting in decrease in plasma phosphate levels in vivo. IP6K inhibition is therapeutically relevant and improves hyperphosphataemia and associated complications. IP6K inositol hexakisphosphate kinase.

Gene knockout models are unlikely to demonstrate the physiological significance of inositol pyrophosphates generated by IP6K, which is attributed to the fact that IP6K regulates target proteins via protein–protein interactions in addition to its enzymatic activity-produced inositol pyrophosphates[2]. Therefore, the combination of SC-919, genetic models, and recently developed inositol pyrophosphate analogues[35] could expand our understanding of inositol pyrophosphate biology.

The experimental use of TNP, the most commonly available IP6K inhibitor, is limited owing to its poor potency, off-target effects, and insufficient in vivo efficacy[36,37]. Medicinal research has reported that even new IP6K inhibitors show limited potency[36,38]. SC-919 was discovered through the optimization of library hit compounds based on potency, physicochemical properties, pharmacokinetic profiles, and safety profiles[18]. The excellent profiles of SC-919 may help overcome the limitations associated with currently available IP6K inhibitors. A limitation of the current study is that the inhibitory activity of SC-919 against kinases closely related to IP6K [inositol 1,4,5-trisphosphate kinase (IP3K; E.C. 2.7.1.127) and inositol polyphosphate multikinase (IPMK; E.C. 2.7.1.151)][39] was not evaluated. However, a recent study that used our patent information[18], which was published while this manuscript was under review, reported the inhibitory potency of SC-919 against IPMK [IPMK/IP6K1 IC$_{50}$ ratio of SC-919 (LI-2124 in their report) = 82][40], indicating

that SC-919 is selective for IP6K. However, further profiling of SC-919 with regard to these parameters both in vitro and in vivo is warranted.

Another limitation of this study is that we did not determine the change in phosphate distribution in each organ of SC-919-treated animals. *XPR1* is expressed in all tissues in humans[41], while IP6K1 and IP6K2 are expressed broadly and IP6K3 is highly expressed in the muscle tissues of mice and humans[42]. Taken together, with the fact that SC-919 is a pan-inhibitor against IP6Ks, all tissues are theoretical targets of this mechanism, and hence should be clarified in future studies.

In summary, the enzymatic activity of IP6K controlled the export of cellular phosphate via an SPX domain of *XPR1*-dependent mechanism, thereby regulated circulating phosphate and intracellular ATP levels in mammals in vivo. The administration of the IP6K inhibitor was highly effective in inducing therapeutic effects on hyperphosphataemia and kidney damage in a CKD model. These results reveal a novel biological mechanism of phosphate regulation in vivo, which may have clinical significance for the treatment of hyperphosphataemia.

## Methods

**Materials**. All reagents were purchased from Wako Pure Chemical Industries or Sigma–Aldrich unless specified otherwise. SC-919 was obtained from SCOHIA Pharma, Inc. Sevelamer was purchased from Chugai Pharmaceutical Company

Limited. For in vitro studies, compounds were dissolved in dimethyl sulfoxide (DMSO), whereas for in vivo studies, they were dissolved in 0.5% (w/v) methyl-cellulose solution.

**Cells**. HEK293 cells were obtained from the European Collection of Authenticated Cell Cultures (85120602). Parental (wild-type, C631) and *XPR1* KO HAP1 cells (HZGHC004238c008) were obtained from Horizon Discovery. All cells were cultured in a humidified atmosphere containing 5% $CO_2$/95% air at 37 °C.

**Animals**. The rats were housed in a room with controlled temperature (23 °C), humidity (55 %), and lighting (12 h light/dark cycle). The rats were allowed free access to a standard laboratory chow diet (CE-2, CLEA Japan Inc.) and tap water. All procedures were performed in accordance with the protocols approved by the Institutional Animal Care and Use Committee at Shonan Health Innovation Park and complied with The Eighth Edition of the Guide for the Care and Use of Laboratory Animals (NRC 2011). Shonan Health Innovation Park is accredited by The Association for Assessment and Accreditation of Laboratory Animal Care International. For rat studies, 0.5% methylcellulose was used as the vehicle. Blood samples used in this study were obtained from the tail veins of the animals unless indicated otherwise.

**Preparation of IP6K1, IP6K2, and IP6K3 enzymes for IP6K activity**. IP6K1 (coding for amino acid residues 1–441, GenBank Accession No. NM_153273) was cloned into the pENTR/D-TOPO vector and expressed in insect cells as a fusion protein with a GST-tag at its N-terminus. IP6K2 (coding for amino acid residues 1–426, GenBank Accession No. NM_016291) was cloned into the pGEX-6P-1 vector and expressed in *Escherichia coli* as a fusion protein with a GST-tag at its N-terminus. IP6K3 (coding for amino acid residues 1–410, GenBank Accession No. NM_054111) was cloned into the pET15b vector and expressed in *E. coli* as a fusion protein with a His-SUMO-tag at its N-terminus.

The recombinant baculovirus for the expression of GST-IP6K1 was constructed from pENTR/GST-IP6K1 as described in the manual for the BaculoDirect C-term Baculovirus Expression System (Thermo Fisher Scientific). The amplified recombinant baculovirus was added to the suspension cultures of Sf9 cells. The cells were incubated at 27 °C for 3 d following baculovirus infection, collected, and then stored at −80 °C until use. The frozen cells expressing GST-IP6K1 were suspended in the lysis buffer (50 mM Tris [pH 8.0], 150 mM NaCl, 1 mM $MgCl_2$, 5 U/mL benzonase, cOmplete™ (Roche), EDTA-free Protease Inhibitor Cocktail), and centrifuged at $15,000 \times g$ for 30 min. The supernatant was applied to a GSTrap 4B column (GE Healthcare Bio-Sciences AB). The column was subsequently washed with a wash buffer (50 mM Tris [pH 8.0], 300 mM NaCl, 1 mM $MgCl_2$, 5% glycerol) and the protein was eluted with a wash buffer containing 20 mM reduced glutathione. GST-IP6K1 was further purified by passing through a HiLoad 26/60 Superdex 200 pg column (GE Healthcare Bio-Sciences AB) equilibrated with Tris-buffered saline, containing 5% glycerol, 1 mM $MgCl_2$, and 1 mM dithiothreitol (DTT). The concentrations of proteins were determined using Coomassie Protein Assay Kits (Thermo Fisher Scientific) using bovine serum albumin (BSA) as the standard.

The IP6K2 expression plasmid was transformed into *E. coli* BL21 (DE3) cells (Nippon Gene). The expression of GST-IP6K2 was induced by the addition of a final concentration of 1 mM IPTG to the bacterial culture, and further incubation was performed for 12 h at 18 °C. *E. coli* cells were harvested using centrifugation at $5000 \times g$ for 20 min and stored at −80 °C until use. The frozen *E. coli* cells expressing GST-IP6K2 were lysed in lysis buffer (50 mM HEPES [pH 7.5], 1 mM DTT, 5 mM EDTA, 5 U/mL benzonase, 0.5 mg/mL lysozyme) via sonication, and the lysate was centrifuged at $15,000 \times g$ for 30 min. The supernatant was loaded onto a GSTrap 4B column. The column was washed with the wash buffer (20 mM HEPES [pH 7.5], 300 mM NaCl, 5% glycerol, 1 mM DTT), and the protein was eluted with a wash buffer containing 10 mM reduced glutathione. The eluate was loaded onto a HiLoad 26/60 Superdex 200 pg column using SDX buffer (20 mM HEPES [pH 7.5], 150 mM NaCl, 5% glycerol, 6 mM $MgCl_2$). The concentrations of proteins were calculated using a mass extinction coefficient of 10 at 280 nm for a 1% (10 mg/mL) protein solution.

The IP6K3 expression plasmid was transformed into *E. coli* BL21 (DE3) cells. The expression of His-SUMO-IP6K3 was induced by the addition of a final concentration of 0.1 mM IPTG to the bacterial culture that was further incubated for 16 h at 18 °C. The *E. coli* cells were harvested by centrifugation at $5000 \times g$ for 20 min and stored at −80 °C until use. The frozen *E. coli* cells expressing His-SUMO-IP6K3 were lysed in lysis buffer (50 mM HEPES [pH 7.5], 150 mM NaCl, 1 mM DTT, 1 mM $MgCl_2$, 20 mM imidazole, 5 U/mL benzonase) via sonication, and the lysate was centrifuged at $15,000 \times g$ for 30 min. The supernatant was loaded onto a Ni-NTA Superflow Cartridge (QIAGEN). The cartridge was washed with the wash buffer (50 mM HEPES [pH 7.5], 300 mM NaCl, 5% glycerol, 1 mM DTT, 1 mM $MgCl_2$) and the proteins were eluted with the same buffer containing 250 mM imidazole. The eluate was loaded onto a HiLoad 26/60 Superdex 200 pg column with SDX buffer (50 mM HEPES [pH 7.5], 150 mM NaCl, 5% glycerol, 1 mM DTT, 1 mM $MgCl_2$). The concentrations of proteins were calculated using a mass extinction coefficient of 10 at 280 nm for a 1% (10 mg/mL) protein solution.

**ADP-Glo kinase assay for IP6K activity**. The compounds were co-incubated with recombinant human IP6K1 (15 nM), human IP6K2 (12 nM), or human IP6K3 (1.5 nM) in the assay buffer containing 20 mM HEPES (pH 7.4), 6 mM $MgCl_2$, 1 mM DTT, and 0.01% Tween 20. Following the addition of ATP and $InsP_6$ at final concentrations of 15 µM and 50 µM, respectively, a kinase reaction was carried out for 2 h at 22–26 °C. Thereafter, the ADP-Glo™ kinase assay (Promega) was performed to determine kinase activity. The raw data were analysed using Prism 7 (GraphPad Software Inc.), and a four-parameter logistic fit equation was used to determine the $IC_{50}$ values.

**Plasma protein binding**. Plasma samples of rats, monkeys, and humans were purchased from Charles River Laboratories Japan, Inc. (Kanagawa, Japan), Shin Nippon Biomedical Laboratories, Ltd. (Kagoshima, Japan), and Tennessee Blood Services (Tennessee, USA), respectively. In vitro plasma protein binding was evaluated by using liquid chromatography-tandem mass spectrometry following ultracentrifugation.

**Synthesis of InsP₇**. $InsP_7$ was synthesized according to a procedure described previously[43].

**Preparation of the D-form of InsP₆ Na salt (InsP₆-d₆)**
*Hexa-O-(3-oxo-1,5-dihydro-3λ⁵-2,4,3-benzodioxaphosphepin-3-yl)-myo-inositol-1,2,3,4,5,6-C-d₆.*
To a mixture of *myo*-inositol-1,2,3,4,5,6-C-d₆ (100.0 mg, 0.537 mmol), 1*H*-Tet-razole (564.3 mg, 8.06 mmol), and $CH_2Cl_2$ (20 mL), *o*-xylene N,N-diethylpho-sphoramidite (0.902 mL, 4.18 mmol) was added at 0 °C. The mixture was then warmed to 22–26 °C, and after stirring it for 5.5 days at 22–26 °C, the mixture was cooled to 0 °C. A solution of *m*-chloroperbenzoic acid (ca. 70 %, 6.62 g, 26.9 mmol) in $CH_2Cl_2$ (50 ml), which was dried over $MgSO_4$, was added to the mixture at 0 °C. The mixture was warmed to 22–26 °C. After stirring the mixture for 4 h at 22–26 °C, 10% $Na_2SO_3$ (aq.) was added to it. Following $CH_2Cl_2$ removal by evaporation, the residue was extracted with EtOAc. The extract was washed with 10% $Na_2SO_3$ (aq.), sat. $NaHCO_3$ (aq.), and brine, dried over $MgSO_4$, and concentrated in vacuo. The residue was purified using preparative HPLC (column: Gemini 5 µM NX-C18 110 LC-column 50 × 21.2 nm, elution: 33–73% 10 mM $NH_4CO_3$ in water/$CH_3CN$). The fractions, including the desired product, were concentrated in vacuo. Water and EtOAc were added to the mixture. The extraction was carried out using EtOAc. The extract was washed with brine, dried over $MgSO_4$, and concentrated in vacuo to obtain hexa-O-(3-oxo-1,5-dihydro-3λ⁵-2,4,3-benzodioxaphosphepin-3-yl)-*myo*-inositol-1,2,3,4,5,6-C-d₆ (653.9 mg, 0.511 mmol, yield: 48 %) as a white solid.

¹H NMR (300 MHz, DMSO-d₆) δ 5.02-5.31 (12H, m), 5.35–5.48 (4H, m), 5.49–5.78 (8H, m), 7.31–7.64 (24H, m); MS (ESI/APCI) [M + H]⁺ 1279.2.

*D-form of InsP₆ Na salt (InsP₆-d₆)*. A mixture of hexa-O-(3-oxo-1,5-dihydro-3λ⁵-2,4,3-benzodioxaphosphepin-3-yl)-*myo*-inositol-1,2,3,4,5,6-C-d₆ (38.9 mg, 0.0304 mmol), 10% palladium on carbon (wetted with 50% water) (50.0 mg), EtOH (10 mL), and $H_2O$ (20 mL) was hydrogenated under a balloon pressure of $H_2$ at 22–26 °C. Following an overnight stirring at 22–26 °C, the catalyst was removed using filtration and washed with MeOH. The filtrate was concentrated in vacuo to obtain a colourless oil. The resin (Dowex 50WX8 hydrogen form, 2.00 g) was washed with water (2 mL, four times), 1 M NaOH (2 mL, four times), and water (2 mL, four times). A solution of the obtained oil in water (2 mL) was charged into the resin. The resin was then washed with water (2 mL, four times). The filtrates were passed through a membrane filter (Millipore Millex-LH, Hydrophilic PTFE 0.45 µm). The filtrate was freeze dried to obtain the D-form of $InsP_6$ Na salt ($InsP_6$-d₆; 23.2 mg, 0.0250 mmol, yield: 82%) as a white solid.

³¹P NMR (162 MHz, D₂O, 85% $H_3PO_4$ in D₂O standard) δ ppm 1.75 (1 P, s), 2.23 (2 P, s), 2.72 (2 P, s), 3.18 (1 P, s).

**Cell and tissue sampling for the determination of InsP₆ and InsP₇ levels**. Cells (293, HAP1 wild-type, and HAP1 *XPR1* KO) were treated with the indicated concentrations (0.001–1 µM) of SC-919 for 4 h. Thereafter, the cells were immediately treated with trypsin/EDTA (25200056, Thermo Fisher Scientific) and 10% FCS-supplemented medium containing similar concentrations of SC-919. The cells were centrifuged ($2000 \times g$, 1 min, 4 °C) and the supernatant was removed. The animals used for in vivo experiments were 8-weeks-old Sprague-Dawley (SD) rats. SC-919 (0.3, 1, 3, and 10 mg/kg) was orally administered, and after 2 h, the rats were anesthetized with isoflurane (1–5% v/v) and sacrificed by exsanguination to collect tissue samples. The cell and tissue samples were placed at −80 °C until required for analysis. The frozen tissue or cell samples were mixed with 3.6% perchloric acid (v/v, prepared in distilled water [318-90105, Nippon Gene]) (prepared using 0.2 and 0.3 mL of 3.6% PCA per cell sample [$1 \times 10^6$ and $4 \times 10^6$ cells for 293 and HAP1 wild-type/*XPR1* KO cells] and 100 mg tissue samples, respectively). The samples kept in ice were immediately homogenized for 2.5–10 min with zirconia beads (5 mm) using a mechanical homogenizer (1100 rpm, 4 °C; Shakemaster, Bio Medical Science) followed by a spin down. Subsequently, one-third (v/v, against PCA) of 30% (w/v) potassium chloride solution was mixed with the homogenized samples with vortexing. Next, samples were centrifuged

$(20,000 \times g$, 5 min, 4 °C) and supernatant was used for subsequent analyses in $InsP_6$ and $InsP_7$ measurements (prepared cell or tissue samples for next step).

### Determination of $InsP_6$ and $InsP_7$ levels in tissues and cells using LC-MS/MS.

Stock solutions were prepared by dissolving accurately weighted $InsP_6$ [099107, Matrix Scientific] or $InsP_7$ in water to yield a concentration of 10 mg/mL for $InsP_6$, 1 mg/mL for $InsP_7$. These stock solutions were further diluted in water to working solutions for calibration standards. Working solutions for calibration curve ranged from 600 to 600,000 ng/mL for $InsP_6$, 3 ng/mL to 3000 ng/mL for $InsP_7$. Rat blank plasma was mixed with 3.6% perchloric acid (v/v, prepared in distilled water) (prepared using 0.3 mL of 3.6% PCA per 0.1 mL rat blank plasma). Subsequently, one-third (v/v, against PCA) of 30% (w/v) potassium chloride solution was mixed with the mixed rat blank plasma with vortexing. Next, the mixtures were centrifuged ($20,000 \times g$, 5 min, 4 °C), and the supernatant (PCA/potassium chloride-treated plasma) was used as blank matrix for preparation of calibration standard, and rat blank plasma was used as a surrogate matrix for tissues and cells. Calibration standards for $InsP_6$ or $InsP_7$ were prepared by adding 10 μL of working solution to 150 μL of PCA/potassium chloride-treated plasma. Samples for the determination of $InsP_6$ and $InsP_7$ levels were prepared by adding 10 μL of water to 150 μL of prepared cell or tissue samples (see the section of "Cell and tissue sampling for the determination of $InsP_6$ and $InsP_7$ levels"). These calibration standards and samples were spiked with internal standard solution (20 μL, 500 ng/mL $InsP_6$-$d_6$ as an internal standard, 2500 ng/mL $N, N, N', N'$-ethylenediaminetetrakis (methylenephosphonic acid) [E0393, Tokyo Chemical Industry]). After mixing, the mixtures were centrifuged ($20,000 \times g$, 5 min, 4 °C), and the supernatant was subjected to ultrafiltration ($15,000 \times g$, 30 min, 4 °C; Amicon Ultra-0.5 Centrifugal Filter Unit, 3 kDa Molecular Weight Cutoff [UFC5003, Merck KGaA]). The filtrates (96 μL) were mixed with 40 μL of water/hexylamine[H0134, Tokyo Chemical Industry]/acetic acid[01021-00, Kanto Chemical] (volume ratio, 44:3:3, respectively), then subjected to LC-MS/MS. Injection volume for determination of $InsP_6$ and $InsP_7$ was 2 μL and 25 μL, respectively.

### LC-MS/MS conditions for determination of $InsP_6$ and $InsP_7$.

The LC-MS/MS system, a high-performance liquid chromatograph (NANOSPACE NASCA2, OSAKA SODA) coupled to a Triple Quad6500+ mass spectrometer (Sciex) was used for analysis. A YMC-Triart C18 metal free column ($2.1 \times 50$ mm inner diameter, 1.9 μm particle size) [TA12SP9-05Q1PTP, YMC Co., Ltd.] was used for chromatographic separation. Mobile phase A was water/hexylamine/acetic acid/100 mM EDTA/100 mM Methylenediphosphonic Acid (MDP) [M0843, Tokyo Chemical Industry] (volume ratio, 1000:2.64:0.8:0.1:0.05, respectively), mobile phase B was acetonitrile[01031-1B, Kanto Chemical], mobile phase C was methanol [25183-2B, Kanto Chemical]. Mobile phase C was delivered by post-column mixing with liquid chromatography eluent. The flow rate of the mobile phase was set to be 1000 μL/min from injection time until 4 min, 500 μL/min over the next 11 min. Column temperature was set to be 40 °C. Wash solvent 1 of methanol/water/1 M ammonium dihydrogenphosphate (volume ratio, 50:50:2, respectively) and wash solvent 2 of water were used as needle wash solution. After analytes were eluted, a 25 μL of 100 mM MDP as wash solvent was injected into LC-MS/MS at 4.0, 5.5, 7.0, 8.5, and 10.0 min. The gradient was as follows (time: % A, %B, %C); 0–0.5 min: 39.2%, 0.8%, 60%; 0.5–1.0 min: 39.2–25.6%, 0.8–14.4%, 60%; 1.0–4.0 min: 25.6–22%, 14.4–18%, 60%;4.0–4.5 min: 44-4%, 36–76%, 20%; 4.5–5.0 min: 4%, 76%, 20%;5.0–5.5 min: 72%, 8%, 20%;5.5–11.5 min: 72–4%, 8–76%, 20%; 11.5–12.0 min: 4%, 76%, 20%; 12.0–15.0 min: 78.4%, 1.6%, 20%. The eluent was directly ionized in the electrospray negative ionization mode and detected by multiple reaction monitoring (MRM) mode on a Triple Quad6500+. Dwell time of $InsP_6$, $InsP_7$ and $InsP_6$-$d_6$ were set to be 295, 45, and 45 msec for determination of $InsP_6$, 20, 345, and 20 msec for determination of $InsP_7$. Ion source conditions were as follows: curtain gas, 20 psi; nebulizer gas, 60 psi; turbo gas, 80 psi; ion spray voltage, −4500 V; heater gas temperature, 500 °C; collision gas, 11. The detailed MRM conditions for each analyte are described in Supplementary Table 7.

### Evaluation of the export of phosphate from 293 cells.

Cells were precultured in Dulbecco's modified Eagle's medium (DMEM) (043-30085, FUJIFILM Wako), and seeded (30,000 cells/well) in 96-well plates and cultured in a medium containing 1 μCi/mL of $KH_2{}^{32}PO_4$ (NEX060, Perkin Elmer Inc.) for 18 h. The culture medium was removed, and cells were washed with 200 μL of phosphate-free DMEM (11971025, Thermo Fisher Scientific). Thereafter, the cells were treated with DMEM containing 1 mM phosphate and DMSO/SC-919 (the concentrations are shown in each Figure) for 4 h. Following treatment, the medium (50 μL) was collected and mixed with MicroScint™-PS (100 μL, 5 min, Perkin Elmer) for the measurement of $^{32}P$ radioactivity. The remaining medium was removed and cells were washed twice with PBS (−) and treated with the M-PER reagent (200 μL, 78501, Thermo Fisher Scientific) for 5 min for the measurement of intracellular $^{32}P$ radioactivity. Radioactivity was measured with a TopCount NXT (PerkinElmer).

### Evaluation of the uptake of phosphate using 293 cells.

Cells were precultured in DMEM and seeded (30,000 cells/well) in 96-well plates and cultured for 18 h. The culture medium was removed, and the cells were washed with 200 μL of phosphate-

free DMEM. Thereafter, the cells were treated with DMEM containing 1 mM phosphate, 1 μCi/mL of $KH_2{}^{32}PO_4$, and DMSO/SC-919 (the concentrations are shown in each figure) for 4 h. The cells were then washed with PBS (−) twice and treated with the M-PER reagent (200 μL) for 5 min for the measurement of intracellular $^{32}P$ radioactivity. Radioactivity was measured with a TopCount NXT.

### Evaluation of the export of phosphate from HAP1 cells.

Cells (wild-type and $XPR1$ KO) were precultured in Iscove's Modified Dulbecco's Medium (IMDM, 12440053, Thermo Fisher Scientific), seeded (80,000 cells/well) in 96-well plates, and cultured in a medium containing 1 μCi/mL of $KH_2{}^{32}PO_4$ for 18 h. The culture medium was removed, and cells were washed with 200 μL of phosphate-free DMEM, and were further treated with DMEM containing 1 mM phosphate and DMSO/SC-919 (the concentrations are shown in each Figure). Following treatment, the medium (50 μL) was collected and mixed with MicroScin-PS (PerkinElmer, 100 μL, 5 min) for the measurement of $^{32}P$ radioactivity. The remaining medium was removed and the cells were washed twice with PBS (−) and treated with the M-PER reagent (200 μL) for 5 min for the measurement of intracellular $^{32}P$ radioactivity. Radioactivity was measured using a TopCount NXT.

### Evaluation of the uptake of phosphate using HAP1 cells.

Cells (wild-type and $XPR1$ KO) were precultured in IMDM and seeded (80,000 cells/well) in 96-well plates and cultured for 18 h. The culture medium was removed, and cells were washed with 200 μL of phosphate-free DMEM, and treated with DMEM containing 1 mM phosphate, 1 μCi/mL of $KH_2{}^{32}PO_4$, and DMSO/SC-919 (the concentrations are shown in each Figure) for 4 h. The cells were then washed with PBS (−) twice and treated with the M-PER reagent (200 μL) for 5 min for the measurement of intracellular $^{32}P$ radioactivity. Radioactivity was measured using a TopCount NXT.

### Determination of the effects of gain and loss-of-function of $XPR1$ on phosphate export in $XPR1$ KO cells.

Cells ($XPR1$ KO HAP1 cells) were seeded (20,000 cells/well) in 96-well plates and cultured in IMDM. Each construct ($eGFP$, $XPR1$, $XPR1\Delta SPX$) was transfected to the cells 7–10 h after cell seeding using the Xfect transfection reagent. After 18 h culture, the culture medium was replaced with fresh medium (IMDM) containing 1 μCi/mL of $KH_2{}^{32}PO_4$ and cultured for 18 h. The culture medium was removed, and cells were washed with 200 μL of phosphate-free DMEM, and treated with DMEM containing 1 mM phosphate and DMSO/SC-919 (1 μM) for 4 h. Following treatment, the medium (50 μL) was collected and mixed with MicroScint™-PS (100 μL, 5 min) for the measurement of $^{32}P$ radioactivity. The remaining medium was removed and the cells were washed twice with PBS (−) and treated with the M-PER reagent (200 μL) for 5 min for the measurement of intracellular $^{32}P$ radioactivity. Radioactivity was measured using a TopCount NXT.

### Construction of the expression vectors.

The whole coding sequences of human $XPR1$ (Q9UBH6-1 in Uniprot), $\Delta SPX$-human $XPR1$, which lacks the N-terminal 1–177 amino acids of the wild-type $XPR1$, and $eGFP$ (eGFP in Uniprot), were subcloned into the pEBMulti-Neo vector (FUJIFILM Wako), which has a CAG promoter to express downstream sequences.

### Transfection experiments.

Plasmid transfection was performed using the Xfect transfection reagent (Z1318N, Takara Bio) according to the manufacturer's instructions. Briefly, plasmid DNA (5 μg) was mixed with the Xfect reaction buffer (100 μL final volume). Thereafter, the Xfect polymer (1.5 μL) was added, and the mixture was incubated for 10 min at 22–26 °C (final mixture). The final mixture (10 μL) was added to the culture medium (100 μL). The cells were cultured for 18 h, and the culture medium was replaced the next day. The cells were further cultured for 24 h and used for further experimentation.

### Single-dose study in normal rats.

The experiment was performed with 8-week-old male SD rats (CLEA Japan, Inc.). Male rats were randomized into different groups based on their body weight and phosphate levels in the plasma ($n = 5$). SC-919 (1, 3, 10 mg/kg) or vehicle was orally administered to the animals. Blood samples were collected from the tail vein at indicated timepoints for the measurement of phosphate levels in the plasma. The concentration of the compound in the plasma was measured at the same timepoints ($n = 3$).

### Single-dose study in normal monkeys.

The experiments were performed at Hamamatsu Pharma Research, Inc. (Hamamatsu, Shizuoka, Japan), which is accredited by the American Association for Accreditation of Laboratory Animal Care. Male cynomolgus monkeys were housed in a room with controlled temperature, humidity, and lighting (12 h light/dark cycle). All monkeys were fed a regular diet (100 g/day, PS-A, Oriental Yeast Co., Ltd.) and had free access to tap water. The experiments were performed when the animals were 3–4 years old. The monkeys were randomized into different groups based on plasma phosphate, body weight (2.6–4.7 kg), and age ($n = 6$). Corn oil (FUJIFILM Wako) was used as a vehicle. SC-919 (0.3 and 3 mg/kg) or vehicle was orally administered to the animals. Blood was collected from the cephalic vein, saphenous vein, or the femoral vein using a disposable syringe and needle at the indicated timepoints for the

measurement of phosphate levels in the plasma. The concentration of the compound in the plasma was measured at the same timepoints ($n = 6$).

**Determination of the phosphate excretion in the faeces and urine of SC-919-treated rats**. The experiment was performed using 8-week-old SD rats (CLEA Japan, Inc.). Male rats were randomly sorted into different groups based on their body weight ($n = 5$). To determine the effect of the sub-chronic dose, SC-919 (10 mg/kg), vehicle, or lanthanum carbonate (1000 mg/kg) was orally administered once a day for 7 days. To determine the single-dose effect, SC-919 was administered once. After a 1 h treatment, $NaH_2{}^{32}PO_4$ solution (11 μCi/2 mL/rat) was administered orally. Following $NaH_2{}^{32}PO_4$ administration, rats were placed in metabolic cages, and faeces and urine were collected up to 72 h later. The compound was administered once a day during the 72 h experiment. Faeces were dissolved in sodium hypochlorite solution (faeces 100 mg/mL of sodium hypochlorite solution, 197-02206, FUJIFILM Wako); the dissolved faeces (1 mL) and urine (0.1 mL) were mixed with Hionic-Fluor (20 or 10 mL) for the measurement of radioactivity using LSC-6100 (Hitachi). The radioactivity (%) in the faecal and urine samples was calculated by setting the radioactivity of the administered $NaH_2{}^{32}PO_4$ to 100 %.

**Sub-chronic dose study in adenine-treated rats**. To establish hyperphosphataemia, 11-week-old normal male SD rats (CLEA Japan, Inc.) were fed a diet (CE-2, CLEA Japan Inc.) containing 0.75 % (w/w) of adenine (010-11513, FUJIFILM Wako) for 2 weeks. Thereafter, 13-week-old rats were randomized into different groups based on the levels of creatinine and phosphate in the plasma ($n = 6$). SC-919 (1 and 10 mg/kg) or vehicle was orally administered once a day. Sevelamer hydrochloride was mixed with the diet (0.6% w/w) and administered to the rats. Blood samples were collected from the tail vein at indicated timepoints after the first (day 1) and seventh (day 7) dose to determine the phosphate levels. The concentrations of SC-919 in plasma were determined in 13-weeks-old adenine-treated rats under similar conditions ($n = 3$).

**Single-dose study in rats with bilateral nephrectomy**. The experiment was performed with 9-week-old male SD rats (CLEA Japan, Inc.). For bilateral nephrectomy, rats were fasted for 16 h and were subcutaneously injected with buprenorphine (0.05 mg/kg, Otsuka). Thirty minutes later, the rats were anesthetized using isoflurane (3% v/v) and kept on thermal pads at 37 °C with isoflurane (2% v/v). The left and right abdomen and the neck were shaved with a clipper. The surgical site was disinfected with a chlorhexidine-containing antiseptic solution, and the aesthetic state was confirmed by the disappearance of the pain reflex. The skin was incised to expose the kidney, and the renal artery and vein were ligated, which was followed by the isolation of the kidney. The kidney on the other side had a similar operation, the skin was sutured, and the abdomen was closed. A sham operation group ($n = 6$), without nephrectomy, was prepared following the same procedure. After completion of the operation, meloxicam (1 mg/kg) was subcutaneously administered once a day during the study. One day after the operation, anaesthesia was introduced using the procedure described above, and blood was collected from the jugular vein. In an oral administration study, rats were randomly divided into different groups based on the levels of phosphate in the plasma and the body weight ($n = 7$ for the nephrectomy group and $n = 6$ for the sham group). SC-919 (0.3 and 1 mg/kg) or vehicle was orally administered to the rats and blood samples were collected at 5 and 24 h post administration from the vein and the abdominal vena cava, respectively, under anaesthesia induced by isoflurane (2–3% v/v) for the determination of plasma parameters.

**Determination of ATP levels in 293 cells**. To determine ATP levels at different timepoints over the time course (up to 4 h), the cells were precultured in DMEM, seeded (20,000 cells/well) in 96-well plates, and cultured for 18 h. Cells were then treated with DMSO or SC-919 at the indicated concentration and intracellular ATP levels were determined at the indicated timepoints. ATP levels in the cells were determined using CellTiter-Glo® Luminescent Cell Viability Assay (Promega) with a standard curve for ATP.

**Determination of ATP Levels in HAP1 wild-type and XPR1 KO Cells**. The intracellular ATP levels in wild-type and XPR1 KO HAP1 cells were determined in cells cultured with IMDM. To compare the effects of SC-919 on the ATP levels in wild-type and XPR1 KO HAP1 cells, each cell type was treated with DMSO or SC-919 at the indicated concentrations for 4 h.

**Determination of the effect of gain and loss-of-function of XPR1 on intracellular ATP levels in XPR1 KO cells**. The XPR1 KO HAP1 cells were seeded (20,000 cells/well) in 96-well plates and cultured in IMDM. Each construct (eGFP, XPR1, XPR1ΔSPX) was introduced after 7–10 h of cell seeding using the Xfect transfection reagent. After 18 h of culture, the culture medium was replaced with fresh medium (IMDM) and cultured for 18 h. The cells were treated with DMEM containing 1 mM phosphate and DMSO/SC-919 (1 μM) for 4 h. Following treatment, the medium was removed, and intracellular ATP levels were determined.

**Cell treatment with SC-919 to evaluate the contribution of intracellular phosphate to ATP synthesis**. 293 cells were precultured in DMEM and then seeded (2,000,000 cells/well) in six-well plates and cultured with DMEM containing 1 mM phosphate and 20 μCi/mL of $KH_2{}^{32}PO_4$ for 18 h. Thereafter, the culture medium was removed, and the cells were washed with 200 μL of phosphate-free DMEM. The cells were subsequently treated with DMEM containing 1 mM phosphate and DMSO/SC-919 for 4 h.

**Cell treatment with SC-919 to evaluate the contribution of extracellular phosphate to ATP synthesis**. 293 cells were precultured in DMEM and then seeded (20,000,000 cells/well) in six-well plates and cultured with DMEM containing 1 mM phosphate for 18 h. The culture medium was subsequently removed, and the cells were washed with 200 μL of phosphate-free DMEM. The cells were treated with DMEM containing 1 mM phosphate, 20 μCi/mL of $KH_2{}^{32}PO_4$, and DMSO/SC-919 for 4 h.

**Detection of phosphate incorporated into ATP using an HPLC-based method**. Following cell treatment with $^{32}P$-labelled phosphate ($KH_2{}^{32}PO_4$), the medium was removed and cells were washed with PBS (−) and dislodged from the culture surface by treatment with trypsin/EDTA, then collected in a tube (15 mL tube) using centrifugation ($1860 \times g$, 5 min). The cells were homogenized in 200 μL of 0.4 M perchloric acid with 5 mm zirconia beads using a mixer (3500 rpm, 5 min, R. T.; model TS-100, Thermal Chemical) followed by a spin down. The homogenized cell solution was resuspended in 490 μL of 2 M potassium carbonate, vortexed, and centrifuged ($20,400 \times g$, 5 min, 4 °C). The supernatant was filtered using the AMICON ULTRA ($15,030 \times g$, 40 min, 4 °C; Amicon Ultra-0.5 Centrifugal Filter Unit, 3 kDa Molecular Weight Cutoff [UFC5003, Merck KGaA]), and the ATP fraction was separated using a Prominence HPLC instrument with a UV/Vis detector (Shimadzu) (injection volume, 15 μL). The conditions used for HPLC were as follows: System: Shimadzu HPLC system (LC-20A, low pressure gradient); Column: Shim-pack XR-ODS III (200 mm L × 2.0 mm I.D., 2.2 μm) [228-59910-92, Shimadzu]; Mobile phase: (A) 100 mmol/L phosphoric acid, 150 mmol/L triethylamine solution, (B) 90/10 (v/v) mixture of mobile phase A/acetonitrile; 99% A and 1% B at 0 min; 99 % A and 1% B at 60 min; stop at 60.01 min; flow rate, 0.3 mL/min; wavelength, 260 nm; injection volume, 30 μL; column temperature, 50 °C. The fractions for ATP measurement were separated using HPLC. Samples were collected shortly after confirmation of the rise in the ATP peak using a UV/Vis detector and was finished 30 sec later for the complete elution of the ATP peak. Radioactivity was measured using a Tri-Carb 3110TR Liquid Scintillation Analyzer (Perkin Elmer.).

**Chronic dose study in adenine-treated rats**. To establish hyperphosphataemia, 11-week-old normal male SD rats (CLEA Japan, Inc.) were fed a diet (CE-2) containing 0.75% (w/w) of adenine for 3 weeks. Thereafter, 14-week-old rats were randomized into groups ($n = 12$ for adenine-treated rats and $n = 6$ for normal rats) based on the levels of plasma creatinine, plasma phosphate, and body weight. SC-919 (1 and 10 mg/kg) or vehicle was dosed once daily for 5 weeks. Rats were fed a standard diet (CE-2) during the study were used as normal control. Sevelamer was dosed as a food admixture (1% w/w). One week after the start of the experiment, the adenine diet was replaced with the standard diet (CE-2), and the rats were maintained for another 4 weeks (total 5 weeks). Blood samples were collected from the tail vein at the indicated timepoints to measure plasma parameters, and food intake was measured regularly. At the end of the study, the rats were anesthetized with isoflurane (1–5% v/v) and were sacrificed to collect blood and tissue samples (aorta and kidney). The aortic arch was sampled from the aortic root, immersed in liquid nitrogen, and stored at −80 °C. To measure calcium content in the aorta, the samples were dried at 105 °C for 4 h using a thermostat (ISUZU VTN-115), weighed, and heated at 600 °C for 1.5 h using a muffle furnace (ADVANTEC KM-420; Advantec) to ash. The incinerated samples were dissolved in nitric acid (FUJIFILM Wako), and the calcium level per tissue dry weight was measured using an ICP emission spectrometer (Thermo Fisher Scientific iCAP6300 Duo). Kidneys were fixed using 10% neutral buffered formalin, paraffin sections were prepared, and then stained with Sirius Red. Images were captured with a NanoZoomer digital slide scanner (Hamamatsu Photonics K.K.) and analysed using WinROOF2015 software (MITANI Corporation.). Ten sites were extracted per section, and the percentage of red stained areas, which is fibrotic, in the tissue area was calculated.

**Measurements of plasma parameters**. The levels of intact PTH, FGF23, 1,25-dihydroxy vitamin D [1,25D], and GDF15 in the plasma were determined using rat intact PTH ELISA kit (60-2500, Immutopics International, San Clemente, CA, USA), FGF23 ELISA kit (CY-4000, KINOS Inc., Tokyo, Japan), 1,25-(OH)₂ Vitamin D EIA (AC-62F1, Immunodiagnostic Systems), and Mouse/Rat GDF15 Quantikine ELISA Kit (MGD150, R&D Systems), respectively. The phosphate levels in the plasma were determined using a Malachite Green Phosphate Assay Kit (POMG-25H, BioAssay Systems) or Autoanalyzer 7180 (Hitachi). The levels of creatinine in the plasma were determined using an Autoanalyzer 7180 (Hitachi).

**Determination of kidney collagen levels**. The kidney samples were incubated in 6 N HCl at 95 °C for 20 h to hydrolyse collagen to hydroxyproline. After centrifugation (13,000 × $g$, 10 min, 20 °C), hydroxyproline content in the supernatants was quantified using the Total Collagen Assay Kit (#QZBTOTCOL1, QuickZyme Biosciences).

**Determination of ATP levels in the kidney**. For establishing hyperphosphataemia, 11-week-old male SD rats (CLEA Japan, Inc.) were fed a diet (CE-2) containing 0.75% (w/w) adenine for 3 weeks. Thereafter, 14-week-old rats were randomly sorted into groups based on the levels of plasma creatinine, plasma phosphate, and body weight ($n = 6$). SC-919 (1 and 10 mg/kg) or vehicle was orally administered once a day. Sevelamer hydrochloride was mixed with the diet (1% w/w) and administered to the rats. After 6 h of the eighth dose of SC-919, rats were sacrificed under isoflurane (3–5%) anaesthesia and kidney samples were collected, frozen, and kept at −80 °C until use. The levels of ATP in the samples were determined using the AMERIC-ATP(T) Kit (FUJIFILM Wako) combined with the CellTiter-Glo® Luminescent Cell Viability Assay kit, and a standard curve for ATP was prepared using rATP (Promega).

**Statistics and reproducibility**. Statistical significance was analysed using Bartlett's test for homogeneity of variances, followed by two-sided Williams' test ($P > 0.05$) and Shirley–Williams test ($P \leq 0.05$) for evaluating the dose-dependent effects and two-sided Dunnett's test ($P > 0.05$) and Steel test ($P \leq 0.05$) for multiple comparisons. Alternatively, statistical significance was analysed using the F test for homogeneity of variances, followed by a two-sided Student's $t$-test ($P > 0.2$) or an Aspin–Welch test ($P \leq 0.2$). All statistical tests were conducted with a two-tailed significance level of 5% (0.05). All data are presented as the mean ± standard deviation. Reliability of the data was ensured by using biological replicates. All attempts at replication of the experimental findings more than one time were reliably reproduced.

**Reporting summary**. Further information on research design is available in the Nature Research Reporting Summary linked to this article.

## Data availability

All data that support the findings of this study are included in the manuscript or are available from the authors upon reasonable request. Reagents presented in this study may be available upon reasonable request under Material Transfer Agreement. Source data are provided with this paper.

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

## Acknowledgements

We thank Kaori Nakanishi for support of experimental work, Yoshito Terao, Masashi Takahashi, Ryuichi Tozawa, Hiroyuki Hozumi, Yoshitaka Yasuhara, Tsuyoshi Ishii, Akito Hata, Masanori Nakakariya for discovering SC-919, Ryuichi Nishigaki, and Yoshinori Satomi for establishing the methods for the measurement of InsP$_6$ and InsP$_7$. The study was financially supported by SCOHIA PHARMA, Inc.

## Author contributions

Y.M., S.A., H.A., A.K., R.K., R.H., S.K. and M.W. conceived and designed the study. Y.M., S.A., H.A., A.K., R.K. R.H., and S.K. conducted the experiments. R.H. and S.K. prepared reagents. Y.M., S.A., H.A., A.K., R.K., R.H., S.K. and M.W. analysed and interpreted the data. Y.M. wrote the manuscript and Y.M., S.A., H.A., A.K., R.K., R.H., S.K. and M.W. reviewed and revised key intellectual contents of the manuscript. Y.M., S.A., H.A., A.K., R.K., R.H., S.K. and M.W. agree to be accountable for all aspects of the work, ensuring that questions related to the accuracy or integrity of any part of the work were appropriately investigated and resolved.

## Competing interests

All authors are employees of SCOHIA PHARMA, Inc. SCOHIA has been granted by Takeda Pharmaceutical Company an exclusive worldwide license to research, develop, manufacture, commercialize, or otherwise exploit SC-919 as a drug candidate for human use under the patent WO2018182051.
