## [Peer Review File · Nature Communications]

REVIEWER COMMENTS

Reviewer #1 (Remarks to the Author):

In this manuscript, the authors have characterized the pan-IP6K inhibitor drug SC-919 in both rats and monkeys obtaining interesting 'real physiology' results. The ability of SC-919 to alleviate hyperphosphataemia and improve chronic kidney disease symptoms is extremely interesting. The authors couple the important and original animal physiological studies with in vitro and in cell characterization of the effects of the drug. The results from the latter set of experiments are overall confirmative of previous literature, even the drug SC-919 was previously described (reference 7). The novelty of this work mostly depends on the animal experiments and on the suggested therapeutically relevance of SC-919. Thus, the work is important and could have a great impact on human health. However, there are several issues that must be addressed to improve the experimental data.

1) In Figure 1C the authors demonstrate that the kidney possesses three times more IP7 than the liver and 50 times more IP7 than the muscle. The kidney, together with the intestine and the bones, is the main organ regulating blood phosphate levels. The fact that the authors did not comment on this elevated level of IP7 in kidneys is somewhat odd. The authors should discuss these differences. Furthermore, the SC-919 effect on kidney IP7 levels is dramatic (higher than 90%), far greater than in the other two tissues analysed. The data require some explanation. Does SC-919 accumulate in the kidney? It is eliminated by the kidney? Or it is metabolized in the kidney? The last point is especially important as kidney cells reabsorb phosphate from the lumen (thanks to the NaPi cotransporter IIa and IIb and to Pit-2) and transfer it through the basolateral membrane to the bloodstream. Thus, in the basolateral membrane XPR1 should export the luminal reabsorbed phosphate to the bloodstream. If SC-919 blocks kidney cells XPR1 phosphate export is also blocking the reabsorption of phosphate from the renal lumen and as a consequence, this should lead to increased phosphates in the urine. However, figure 3d demonstrates exactly the opposite. How do the authors explain these data? Given the importance of the kidneys in maintaining phosphate homeostasis and the dramatic effect of SC-919 on normally elevated kidney IP7 levels, the authors must carefully consider how the drug, IP7, and XPR1 work in kidney cells.

2) Cloning, cell manipulation, and animal work are described in great detail in the Methods section in the supplementary materials. Unfortunately, how the authors measured IP6 and IP7 levels is not explained. The authors developed a new LC-MS/MS but do not show any validation chromatograms. The authors must validate their new methods giving more technical details and especially confirmatory data. They must compare their new LC-MS/MS protocols with the recently published HILIC-MS/MS method (PMID: 30220429). What is plaguing IP7 research is the lack of reliable analytical methods. The authors must convince this reviewer that they are actually measuring IP6 and IP7.

Similarly, the section on IP7 and IP6-d6 organic synthesis lacks validation data. The NMR analysis are reported only as numbers and the actual spectra are not show. How clean are these NMR spectra? Showing the NMR spectra is standard in organic synthesis. NMR spectra do not give accurate purity assessment, thus the authors should run a PAGE analysis (PMID: 19440344) of their newly synthesized IP6-d6 and IP7 to assess the purity.

3) The introduction could be expanded, currently, it is very short. IP7 and phosphate homeostasis is now a very hot topic. However, ten years ago how inositol pyrophosphates control cellular phosphate homeostasis was not considered a worthy line of investigation.

4) In the result section, while presenting some of the data in figures 2 and 3, reference 17 must be cited since the data are very similar.

5) The authors should improve figures quality. Often the X-axes of panels on the same line are not even aligned. Fig 2B left and right panel, Fig 2 F and G, Fig 3C and D, and many more.

6) Lines 60, 157, and 158 IPK6 was used instead of IP6K

Reviewer #2 (Remarks to the Author):

Moritoh and colleagues have demonstrated that IP6K is required for phosphate regulation in vivo, through a mechanism involving altered phosphate export and intracellular ATP levels. The authors also have showed that this pathway can be targeted in the disease setting of chronic kidney disease.

Overall this study elucidates a novel pathway, through a series of well designed in vitro and in vivo experiments, and highlights a new potential therapeutic strategy. I have the following comments:

The title puts this in the context of the mechanism in mammals which implies potentially numerous different species. On a similar theme, it would be useful to read a more wider discussion of the evolutionary context / benefits of this mechanism.

A comparison of circulating pyrophosphate (ppi) levels, and regulators of PPI (eg Enpp1, Ank, Abcc6) would have been useful to fully explore the phosphate axis.

Where FGF23 levels altered following treatment of the CKD model.

A stronger justification for using monkeys would be helpful.

formatting of figures - would be easier to read if each individual graph was labelled A,B,C etc

A more detailed characterisation of the vascular calcification changes would be useful eg histology, imaging

Reviewer #3 (Remarks to the Author):

The enzymatic activity of IP6K controls circulating phosphate in mammals

This is a well conducted study regarding a relevant issue. CKD patients should greatly benefit from convenient treatment of hyperphosphatemia since lowering serum phosphate in these patients should result in a reduced cardiovascular co-morbidity/mortality. Efficiency of current therapies for hyperphosphatemia (mainly intestinal phosphate binding) is compromised by relatively low therapy-compliance due to gastro-intestinal side-effects.

The reviewer however has some concerns

1. The introduction should benefit from figures of (1) chemical structures and (2) pathways inositol pyrophosphate metabolism
2. Fig 2F: I should expect the extracellular exported phosphate in XPR1 knockout cells to be at least as low as in the sc-919 treated cells, please explain.
3. Fig 2F-G: the authors should clearly explain what is the meaning of the different study set-ups used in F and G, and why these different set-ups were used.
4. Fig 2 H-I: X-axis legend is not clear. Authors have to explain I (why intracellular phosphate is the same in XPR1 and XPR1/deltaSPX cells).
5. Fig 3: how SC919 was administered in rats (for monkeys is mentioned orally)
6. Fig 3 D: y-axis scale not appropriate
7. Why lanthanum carbonate is used as a control in the experiments of Figure 3 (normal rats) and sevelamer in the experiments of Figure 4 (hyperphosphatemic rats).
8. The authors have to clarify why serum creatinine/phosphate is decreasing over time in vehicle treated animals
9. Next to plasma phosphate, also plasma creatinine levels are significantly lower in sc-919 treated rats. Since plasma creatinine levels reflect renal function of the animals it is perfectly possible that the lower plasma phosphate levels in these animals are the result of a better kidney function and not from an altered cellular phosphate metabolism.

10. Since my concern in 9, the authors have to add information regarding the target organ of sc-919. Is phosphate excretion inhibited in all cells of the body (for example: liver, vessels, kidney). In which organs XPR1 is expressed?

11. Why only ATP is measured in kidney tissue. It can be both the reflection of a better kidney function and/or an altered cellular phosphate mechanism. Therefore ATP has to be measured in other organs to.

Reviewer #4 (Remarks to the Author):

The manuscript entitled "The enzymatic control of IP6K controls circulating phosphate in mammals" describes the application of a small-molecule inhibitor targeted against IP6Ks. The authors demonstrate that in vivo inhibition of IP6Ks decreased plasma phosphate levels, which was correlated with a decrease in intracellular PP-InsP levels. Chronic IP6K inhibition proved effective to alleviate hyperphosphatemia and improved kidney function in chronic kidney disease. Given that so far no potent and selective IP6K inhibitor has been available, this study marks a significant contribution to the field. However, since everything hinges on the inhibitor (its characterization, its properties, its influence on cellular PP-InsP levels), this portion of the manuscript will need additional work before the paper would be suitable for publication in Nature Communications.

Major points:

- The introduction is too short and does not provide the necessary context for the work. The authors are not the first to discover a connection between PP-InsPs and phosphate homeostasis. There is a significant body of work in other organisms, such as yeast and plants, that clearly shows a regulatory role for PP-InsPs in phosphate homeostasis. Also in mammalian systems, GWAS have linked IP6K3 to plasma phosphate levels, and phosphate export from mammalian cells was demonstrated to be dependent on IP6Ks/PPIP5Ks (the latter two studies were cited by the authors).
- Given that the authors are able to provide such a great pharmacological tool, they need to mention and cite the currently available IP6K inhibitors (TNP, TNP analogs, flavonoids etc.)
- Compound SC-919 was discovered in a HTS, followed by compound optimization. More information on the screen is required: What were the screening conditions, number of compounds screened, Z'-values, analysis parameters and counter screen conditions? How did the authors deal with the inherent ATPase activity of IP6Ks, which results in the detection of false positive hits?
- Full characterization of SC-919 is missing. This compound contains two stereocenters. Were the isomers separated? Do they show the same potency? Or was a mixture used? The ¹H and ¹³C NMR data need to be reported, as well as HRMS data. This compound was used in all subsequent experiments, so this information is critical.
- Figure 1b is not as informative as it could be. The assay conditions should be noted in the caption. But more importantly, the assay conditions for the different IP6Ks were not identical (different protein concentrations were used), impairing comparability of IC₅₀ values. Moreover, an ATP concentration of 50 μmol is very low (and physiological irrelevant), and an odd choice, given the kinetic properties of IP6Ks. I am assuming the authors checked whether SC-919 is indeed ATP competitive?
- The kinase panel (table S1) looks promising, and highlights the good selectivity of SC-919 towards the IP6Ks. Nevertheless, several protein kinases were partially inhibited at 1 μM concentration. While this concentration appears high (when compared to the biochemical assays), the doses used in the animal studies are not low. As can be seen in Figure 3A, the actual inhibitor concentration in plasma after treatment is more than 10 μM, and hence significantly higher than the amount used to screen the kinase panel. Also, at 10 μM some off-target effects are to be expected. The authors should mention this limitation in the text. Furthermore, the kinase panel was mainly composed of protein kinases. While it was useful to include the lipid kinases (PI3Ks), it would also be informative to know the potency of SC-919 against the most closely related small molecule kinases IPMK and ITPK1. Have these kinases been tested?
- I was intrigued to learn about the LC-MS/MS method to quantify cellular InsP6 and InsP7 levels and the preparation of the deuterated InsP6 standard presents another useful tool. However, I

could not follow how this method really worked based on the information provided. Significantly more details need to be provided here, as the method is used throughout the paper as a quantification method. Details about column, ion mode, calibration, and limit of detection need to be provided. How can 5PP-InsP5 be quantified without internal standard, and how can 5PP-InsP5 be distinguished from 1PP-InsP5? Is InsP8 detectable? And InsP5?

- I am missing a discussion about the different mechanistic possibilities towards the end.

Reference 3 and reference 5 have proposed 5PP-InsP5 and InsP8 as critical regulators in phosphate export. In the opinion of the authors, are both inositol pyrophosphates relevant? What has the current study contributed to this mechanism? The manuscript confirmed the necessity of the SPX domain on XPR1 to observe the effects of SC-919, but are there certain conclusions that can be drawn from the use of this new pharmacological tool? How could the pharmacological tool be used in combination with genetics to address the specificity issues in PP-InsP signaling? In my opinion, there is a lot of potential, and the authors should discuss this more carefully.

Minor comments:

- Line 157 and 158. IP6K instead of IPK6.

- Line 213 comma after IP6K

- Supporting Information Page 8: Synthesis of InsP7: I couldn't find the reference for the synthesis

REVIEWER COMMENTS

Our responses to the reviewers' comments

Reviewer #1 (Remarks to the Author):

In this manuscript, the authors have characterized the pan-IP6K inhibitor drug SC-919 in both rats and monkeys obtaining interesting 'real physiology' results. The ability of SC-919 to alleviate hyperphosphataemia and improve chronic kidney disease symptoms is extremely interesting. The authors couple the important and original animal physiological studies with in vitro and in cell characterization of the effects of the drug. The results from the latter set of experiments are overall confirmative of previous literature, even the drug SC-919 was previously described (reference 7). The novelty of this work mostly depends on the animal experiments and on the suggested therapeutically relevance of SC-919. Thus, the work is important and could have a great impact on human health. However, there are several issues that must be addressed to improve the experimental data.

1) In Figure 1C the authors demonstrate that the kidney possesses three times more IP7 than the liver and 50 times more IP7 than the muscle. The kidney, together with the intestine and the bones, is the main organ regulating blood phosphate levels. The fact that the authors did not comment on this elevated level of IP7 in kidneys is somewhat odd. The authors should discuss these differences. Furthermore, the SC-919 effect on kidney IP7 levels is dramatic (higher than 90%), far greater than in the other two tissues analysed. The data require some explanation. Does SC-919 accumulate in the kidney? It is eliminated by the kidney? Or it is metabolized in the kidney? The last point is especially important as kidney cells reabsorb phosphate from the lumen (thanks to the NaPi cotransporter IIa and IIb and to Pit-2) and transfer it through the basolateral membrane to the bloodstream. Thus, in the basolateral membrane XPR1 should export the luminal reabsorbed phosphate to the bloodstream.

If SC-919 blocks kidney cells XPR1 phosphate export is also blocking the reabsorption of phosphate from the renal lumen and as a consequence, this should lead to increased phosphates in the urine. However, figure 3d demonstrates exactly the opposite. How do the authors explain these data? Given the importance of the kidneys in maintaining phosphate homeostasis and the dramatic effect of SC-919 on normally elevated kidney IP7 levels, the authors must carefully consider how the drug, IP7, and XPR1 work in kidney cells.

Response

Thank you for your comment and assessment. We observed unexpectedly high levels of

InsP7 in the kidney, and therefore we have added further discussion regarding InsP7 levels in the revision manuscript (page 13, lines 278-283). The absolute InsP7 levels across organs have not been fully investigated owing to the limitations associated with the available methodology, and hence we are interested in determining the *in vivo* InsP7 levels. As indicated by the reviewer, the effect of SC-919 on the kidney was stronger than that on the liver and muscles. We measured the tissue levels of SC-919 following its oral administration and observed higher levels of SC-919 in the kidney, which may potentially decrease the InsP7 levels (page 5, lines 88-89 and Table S5). With respect to the urine excretion of phosphate, XPR1 in the kidney has a physiological role with respect to phosphate reabsorption. Phosphate is eliminated through the glomerulus, depending on its plasma concentration, and when the renal excretion of phosphate is reduced through multiple mechanisms, phosphate excretion in urine is decreased. In the present study, SC-919 was administered to normal rats which decreased plasma phosphate levels below normal range. Therefore, decreased plasma phosphate concentrations are likely a factor for the decreased urine phosphate excretion in SC-919-treated rats. To elucidate the role of IP6K-regulated XPR1 on kidney phosphate reabsorption, a clamp study using animals with normal plasma phosphate levels is essential.

2) Cloning, cell manipulation, and animal work are described in great detail in the Methods section in the supplementary materials. Unfortunately, how the authors measured IP6 and IP7 levels is not explained. The authors developed a new LC-MS/MS but do not show any validation chromatograms. The authors must validate their new methods giving more technical details and especially confirmatory data. They must compare their new LC-MS/MS protocols with the recently published HILIC-MS/MS method (PMID: 30220429). What is plaguing IP7 research is the lack of reliable analytical methods. The authors must convince this reviewer that they are actually measuring IP6 and IP7. Similarly, the section on IP7 and IP6-d6 organic synthesis lacks validation data. The NMR analysis are reported only as numbers and the actual spectra are not shown. How clean are these NMR spectra? Showing the NMR spectra is standard in organic synthesis. NMR spectra do not give accurate purity assessment, thus the authors should run a PAGE analysis (PMID: 19440344) of their newly synthesized IP6-d6 and IP7 to assess the purity.

Response

In the revised manuscript, we have included a detailed method and validation data for our method (pages 18-20, 21-23, supplemental information page 11-13, table S2, table s3, table s4, figure s3, figure s4). As suggested, we also compared our method and those previously published (page 13, lines 275-278). 5-InsP7 was synthesized using the method described in Nat Commun. 2016;7:10622. doi: 10.1038/ncomms10622, which was cited in the revised manuscript (page 18, lines 396-397). NMR spectra of InsP7 and InsP6-d6 are also included which show good purity of both compounds (Supplemental information, pages 10, Figure s2). We also performed PAGE analysis and demonstrated the purity of InsP7 (Supplemental information, pages 10).

3) The introduction could be expanded, currently, it is very short. IP7 and phosphate homeostasis is now a very hot topic. However, ten years ago how inositol pyrophosphates control cellular phosphate homeostasis was not considered a worthy line of investigation.

Response

The introduction was expanded to cover the general background regarding the biology of inositol pyrophosphates and phosphate homeostasis. (pages 2-4)

4) In the result section, while presenting some of the data in figures 2 and 3, reference 17 must be cited since the data are very similar.

Response

Reference 17 was cited in the result section (page 5, line 101).

5) The authors should improve figures quality. Often the X-axes of panels on the same line are not even aligned. Fig 2B left and right panel, Fig 2 F and G, Fig 3C and D, and many more.

Response

We apologise for this inconvenience. The Figure quality has been improved.

6) Lines 60, 157, and 158 IPK6 was used instead of IP6K

Response

Thank you for your suggestion and we apologise for these typos, which have been corrected.

Reviewer #2 (Remarks to the Author):

Moritoh and colleagues have demonstrated that IP6K is required for phosphate regulation *in vivo*, through a mechanism involving altered phosphate export and intracellular ATP levels. The authors also have showed that this pathway can be targeted in the disease setting of chronic kidney disease.

Overall this study elucidates a novel pathway, through a series of well designed *in vitro* and *in vivo* experiments, and highlights a new potential therapeutic strategy. I have the following comments:

The title puts this in the context of the mechanism in mammals which implies potentially numerous different species. On a similar theme, it would be useful to read a more wider discussion of the evolutionary context / benefits of this mechanism.

Response

Thank you for your comment. The fundamental connection between inositol pyrophosphates and phosphate regulation is likely preserved across plants, yeasts, and mammals. A more detailed background of inositol pyrophosphate biology has been added to the revised introduction (pages 2-4). We demonstrated the therapeutic effect of IP6K inhibition even on impaired kidney function without unfavourable gut effects. This mechanism furthermore improved the kidney function parameters. We have added this to the revised discussion (pages 10-11, lines 220-234).

A comparison of circulating pyrophosphate (ppi) levels, and regulators of PPI (eg Enpp1, Ank, Abcc6) would have been useful to fully explore the phosphate axis.

Response

We have not measured the plasma levels of pyrophosphate and molecules that regulate the levels of circulating pyrophosphate, including Enpp1. We believe that each intracellular inositol pyrophosphate and circulating pyrophosphate plays a different physiological role in regulating phosphate levels. However, exploring the relationship between IP6K and circulating pyrophosphate and regulators should be further studied to better understand phosphate regulation *in vivo*. To the best of our knowledge, the relationship between inositol pyrophosphate levels and circulating pyrophosphate levels has not yet been studied, which

is a topic for future research.

Where FGF23 levels altered following treatment of the CKD model.

Response

FGF23 is secreted by bone tissue in response to circulating phosphate levels in CKD (Int J Mol Sci. 2020;21(22):8810. doi: 10.3390/ijms21228810.). However, identifying FGF23-secreting organs influenced by IP6K inhibition remains warranted.

A stronger justification for using monkeys would be helpful.

Response

Thank you for your suggestion. We have included justification for using monkeys in the revised manuscript.

(pages 6-7, lines 131-134)

formatting of figures - would be easier to read if each individual graph was labelled A,B,C etc

Response

Each individual graph has been labelled as suggested.

A more detailed characterisation of the vascular calcification changes would be useful eg histology, imaging

Response

We included representative vascular images (Fig. 6I). As an additional image analysis, we collected kidney samples in the same experiment while the images collected for each group and fibrosis were evaluated. We have included this in the revised manuscript (Fig.6o, p, q, r).

Reviewer #3 (Remarks to the Author):

The enzymatic activity of IP6K controls circulating phosphate in mammals

This is a well conducted study regarding a relevant issue. CKD patients should greatly benefit from convenient treatment of hyperphosphatemia since lowering serum phosphate in these patients should result in a reduced cardiovascular co-morbidity/mortality. Efficiency of current

therapies for hyperphosphatemia (mainly intestinal phosphate binding) is compromised by relatively low therapy-compliance due to gastro-intestinal side-effects.

The reviewer however has some concerns

1. The introduction should benefit from figures of (1) chemical structures and (2) pathways inositol pyrophosphate metabolism

Response

The introduction included the chemical structures and pathways of inositol pyrophosphate metabolism (Fig. 1a, b). These changes were added to the revised manuscript (pages 2-3, lines 36-45).

2. Fig 2F: I should expect the extracellular exported phosphate in XPR1 knockout cells to be at least as low as in the sc-919 treated cells, please explain.

Response

Thank you for your comment. This area is a subject for a further study. We also expected that XPR1 KO cells show lower baseline levels of ³²P-labelled phosphate export; however, we observed similar levels of exported phosphate between wild-type and XPR1 KO cells. These experimental results are highly reproducible. Based on these observations, we speculate that mammalian cells have other mechanism(s) for exporting phosphate in addition to XPR1, which remain to be unidentified. Collectively, we speculate that these unidentified mechanisms may compensate for cellular phosphate export in XPR1 KO cells. However, the mechanism remains to be identified in mammals.

3. Fig 2F-G: the authors should clearly explain what is the meaning of the different study set-ups used in F and G, and why these different set-ups were used.

Response

Thank you for your suggestion. We have included a detailed explanation in the revised manuscript (Figure 2b, c, d, g, h, i, j, k). Briefly Fig. 2b, c, g, h, j and k show the testing phosphate export while 2d and i show the testing phosphate uptake.

4. Fig 2 H-I: X-axis legend is not clear. Authors have to explain I (why intracellular phosphate is the same in XPR1 and XPR1/deltaSPX cells).

Response

We apologise for the inconvenience. The Figure and text have been revised to include more information regarding the experimental conditions (page 6, lines 117-125). Introduction of wild or modified XPR1 that could export phosphate should decrease the intracellular ³²P activity (Fig. 2k). SPX is the N-terminal intracellular domain of XPR1, while the transmembrane domains remained unmodified in the SPX-deleted XPR1. As a result, SPX-deleted XPR1 (deltaSPX) retained the ability to export phosphate as illustrated by Figure 2j. This explains why the intracellular ³²P level remained the same between these treatment groups (Fig. 2k). The detailed explanation has been added to the revised manuscript (page 6, lines 117-125).

5. Fig 3: how SC919 was administered in rats (for monkeys is mentioned orally)

Response

S-919 was administered orally. We have included this information in the revised manuscript (page 6, line 129).

6. Fig 3 D: y-axis scale not appropriate

Response

We have revised the Y axis from 0 to 20%, to make the trend of the various curves more visible (Fig.3g).

7. Why lanthanum carbonate is used as a control in the experiments of Figure 3 (normal rats) and sevelamer in the experiments of Figure 4 (hyperphosphatemic rats).

Response

For a single dose study, a phosphate binder was used as a control drug which can increase phosphate excretion in faeces and decrease urine phosphate excretion. Based on previous results with a similar study design (*Ren Fail.* 2011;33(2):217-24. doi: 10.3109/0886022X.2011.552821.), lanthanum carbonate was more effective than sevelamer at increasing phosphate excretion in faeces. Hence, we used lanthanum carbonate as a control drug in the short period study. Conversely, during the chronic study, to evaluate the effects of SC-919 on phosphate and related complications, a phosphate binder was used as a control drug to lower the levels of circulating phosphate. A previous study in rats with a

similar design (*Kidney Int.* 2003;64(2):441-50. doi: 10.1046/j.1523-1755.2003.00126.x) revealed that sevelamer was effective at lowering the plasma phosphate levels. The rationale behind the use of each agent was that the experimental conditions were mostly known from previous reports.

8. The authors have to clarify why serum creatinine/phosphate is decreasing over time in vehicle treated animals

Response

For enabling a better understanding of the animal model, we included a brief protocol of this experiment in Figure 6a. In this experiment, adenine diet was provided for 3 weeks prior to drug treatment. Following the initial drug dose, adenine diet was provided for an additional week, and was changed to normal diet. Previous studies have shown that continuous administration of the adenine diet results in severe effects in rats; a similar method has previously been used (*Kidney Int.* 2003;64(2):441-50. doi: 10.1046/j.1523-1755.2003.00126.). This explains the decrease in serum creatinine/phosphate levels over time.

9. Next to plasma phosphate, also plasma creatinine levels are significantly lower in sc-919 treated rats. Since plasma creatinine levels reflect renal function of the animals it is perfectly possible that the lower plasma phosphate levels in these animals are the result of a better kidney function and not from an altered cellular phosphate metabolism.

Response

Thank you for your comment. Indeed, decreased plasma creatinine levels in SC-919-treated CKD rats is an indication of better kidney function. We have added this information in the revised manuscript. (pages 9-10, lines 202-203; pages 12-13, lines 269-273).

10. Since my concern in 9, the authors have to add information regarding the target organ of sc-919. Is phosphate excretion inhibited in all cells of the body (for example: liver, vessels, kidney). In which organs XPR1 is expressed?

Response

Thank you for your questions. We have included this information in the revised manuscript. A previous study has shown that XPR1 is expressed in all organs (reference included in pages 13 line 293). IP6K1 and IP6K2 are expressed in all tissues, while IP6K3 is highly expressed in muscles (pages 13-14, lines 293-297). Thus, theoretically, all organs may be potential targets (pages 13-14, lines 293-297). In SC-919-dosed condition, intracellular InsP7 levels could be determined using multiple factors including protein expression levels regulating InsP7, pharmacokinetic profiles of SC-919, and intracellular substrate/metabolic status, *in vivo*. We focused on understanding the physiological consequences by SC-919-mediated IP6K inhibition, and hence further investigations are warranted to address this question (page 14, line 295-297).

11. Why only ATP is measured in kidney tissue. It can be both the reflection of a better kidney function and/or an altered cellular phosphate mechanism. Therefore ATP has to be measured in other organs to.

Response

The current study focused on kidney impairment-driven hyperphosphatemia and associated complications with a clinical relevance. As in this condition, the kidney is likely to be the most relevant organ, we measured the ATP levels in the kidney, whose dysfunction results in the development of hyperphosphatemia in the current model. We speculate that IP6K downregulation-induced inhibition of XPR1 results in elevation of cellular ATP levels, which has protective role in the kidney during CKD (page 12, lines 259-266. However, as suggested, measurement of ATP levels in other organs can also be beneficial, which is a subject for future investigations.

Reviewer #4 (Remarks to the Author):

The manuscript entitled "The enzymatic control of IP6K controls circulating phosphate in mammals" describes the application of a small-molecule inhibitor targeted against IP6Ks. The authors demonstrate that *in vivo* inhibition of IP6Ks decreased plasma phosphate levels, which was correlated with a decrease in intracellular PP-InsP levels. Chronic IP6K inhibition

proved effective to alleviate hyperphosphatemia and improved kidney function in chronic kidney disease.

Given that so far no potent and selective IP6K inhibitor has been available, this study marks a significant contribution to the field. However, since everything hinges on the inhibitor (its characterization, its properties, its influence on cellular PP-InsP levels), this portion of the manuscript will need additional work before the paper would be suitable for publication in Nature Communications.

Major points:

- The introduction is too short and does not provide the necessary context for the work. The authors are not the first to discover a connection between PP-InsPs and phosphate homeostasis. There is a significant body of work in other organisms, such as yeast and plants, that clearly shows a regulatory role for PP-InsPs in phosphate homeostasis. Also in mammalian systems, GWAS have linked IP6K3 to plasma phosphate levels, and phosphate export from mammalian cells was demonstrated to be dependent on IP6Ks/PPIP5Ks (the latter two studies were cited by the authors).

Response

The introduction was revised to include the general background of phosphate regulation among species (page 2-4).

- Given that the authors are able to provide such a great pharmacological tool, they need to mention and cite the currently available IP6K inhibitors (TNP, TNP analogs, flavonoids etc.)

Response

We additionally measured the inhibitory activity of TNP using our assay system and included the result in Table 1 (page 39). All IP6K inhibitors have inhibitory activity comparable with that of TNP, therefore we believe this information is useful. In addition, a reference for TNP was cited to provide information regarding the currently available inhibitors (page 4, line 77).

- Compound SC-919 was discovered in a HTS, followed by compound optimization. More information on the screen is required: What were the screening conditions, number of compounds screened, Z'-values, analysis parameters and counter screen conditions? How did the authors deal with the inherent ATPase activity of IP6Ks, which results in the detection of false positive hits?

Response

The detailed screen condition is not included in the current manuscript since it is out of our study scope, and will be included in another manuscript. The brief information is available in the cited patent (WO2018182051 IP6K INHIBITORS. (2018)). Collectively, we aimed to demonstrate the effects of pharmacological inhibition of IP6K on the body phosphate control, and we believe that the data included in this revised manuscript confirm that SC-919 is an effective inhibitor for IP6K *in vivo*.

- Full characterization of SC-919 is missing. This compound contains two stereocenters. Were the isomers separated? Do they show the same potency? Or was a mixture used? The ¹H and ¹³C NMR data need to be reported, as well as HRMS data. This compound was used in all subsequent experiments, so this information is critical.

Response

SC-919, a single isomer, was prepared stereo-selectively using our developed method (Terao, Y., et al. WO2018182051 IP6K INHIBITORS. (2018)) (Supplemental information, page 2). The absolute configuration of SC-919 was determined using X-ray crystallography (CCDC registration in progress (Deposition Number) 2075338) (Supplemental information, page2 lines 19-21). Analytical data was included in the revision (Supplemental information, Fig.S1).

- Figure 1b is not as informative as it could be. The assay conditions should be noted in the caption. But more importantly, the assay conditions for the different IP6Ks were not identical (different protein concentrations were used), impairing comparability of IC₅₀ values. Moreover, an ATP concentration of 50 μmol is very low (and physiological irrelevant), and an odd choice, given the kinetic properties of IP6Ks. I am assuming the authors checked whether SC-919 is indeed ATP competitive?

Response

We agree that different enzyme concentrations affect the IC₅₀ value. We attempted to adjust the protein concentrations of IP6K1, IP6K2, and IP6K3 to better compare the inhibitory activity of SC-919 against each subtype. However, we were unable to establish the optimised high sensitivity assay for SC-919, although we are still trying. We believe that the inhibitory activity of SC-919 toward IP6K is highly potent and beyond the limits of our assay system. Based on your suggestion, we changed the inhibitory activity of SC-919 of "5.2 nM" to < 5.2 nM for IP6K1, and "3.8 nM" to < 3.8 nM for IP6K2 as these IC₅₀ values are nearly half the enzyme concentration (Table 1). We also tried to elevate the ATP levels to a physiologically

relevant level, but this was difficult due to the background signal of ATP-derived ADP in ADP-glo assay. Thus, we used 15 $\mu\text{mol/L}$ ATP for this assay. In agreement with our assay condition, in a recent paper (J Med Chem. 2019 Feb 14; 62(3): 1443–1454), in which IP6K inhibitors were screened, Shears et al. uses 10 $\mu\text{mol/L}$ ATP for enzymatic assay of human IP6K2 detecting ADP. As described earlier, at the moment, we have no access to other assays such as RI-labelled ATP assay. Therefore, it is difficult to confirm the ATP competitiveness of this compound using this assay system due to limitation of ATP concentration. As mentioned above, we are currently trying to measure the compound activity more accurately and inhibitory mode, however, this may take considerable amount of time. In this manuscript, we focused on the physiological role of IP6K in circulating phosphate levels, *in vivo*.

- The kinase panel (table S1) looks promising, and highlights the good selectivity of SC-919 towards the IP6Ks. Nevertheless, several protein kinases were partially inhibited at 1 μM concentration. While this concentration appears high (when compared to the biochemical assays), the doses used in the animal studies are not low. As can be seen in Figure 3A, the actual inhibitor concentration in plasma after treatment is more than 10 μM , and hence significantly higher than the amount used to screen the kinase panel. Also, at 10 μM some off-target effects are to be expected. The authors should mention this limitation in the text. Furthermore, the kinase panel was mainly composed of protein kinases. While it was useful to include the lipid kinases (PI3Ks), it would also be informative to know the potency of SC-919 against the most closely related small molecule kinases IPMK and ITPK1. Have these kinases been tested?

Response

SC-919 shows relatively high compound exposure to plasma when administered to rats. We performed protein (samples from rats, monkeys, and human) binding analysis for SC-919 and included the results in Table 2, which demonstrated that the free form of SC-919 is approximately 0.1% and 0.8-1.0% of the total compound in rat and monkey plasma, respectively (page 40). Thus, free SC-919 levels are below 0.05 μM in animal studies using rats and monkeys. We do not have access to IPMK and ITPK1 and have not tested the potency of SC-919 against these enzymes. However, as suggested, investigating the closely related kinases should be conducted in collaborations of other researchers in this field. We have included safety information regarding the chronic study in the revised manuscript (page 9, lines 187-188).

- I was intrigued to learn about the LC-MS/MS method to quantify cellular InsP6 and InsP7

levels and the preparation of the deuterated InsP6 standard presents another useful tool. However, I could not follow how this method really worked based on the information provided. Significantly more details need to be provided here, as the method is used throughout the paper as a quantification method. Details about column, ion mode, calibration, and limit of detection need to be provided. How can 5PP-InsP5 be quantified without internal standard, and how can 5PP-InsP5 be distinguished from 1PP-InsP5? Is InsP8 detectable? And InsP5?

Response

Thank you for your comment and questions. The method used to synthesize the d-form of InsP6 is included in the text (pages 18-20, lines 399-434). Also, the method for InsP7 synthesis is included in the methods section of the revised manuscript (page 18, lines 396-397). Details on the column, ion mode, calibration, and limit of detection are included to the revised manuscript as well (pages 21-23, lines 457-506; supplemental table 7). In response to the reviewer's comment, the current method cannot distinguish between 1-InsP7 and 5-InsP7; this information has been added to the revised text (page 4, lines 81-84). Based on the preliminary observations, we may be able to quantify InsP5 and InsP8; however, at this moment, we only validated the quantification methods for InsP6 and InsP7. We are interested in establishing absolute quantification of other inositol phosphates, which requires collaboration with other researchers.

- I am missing a discussion about the different mechanistic possibilities towards the end. Reference 3 and reference 5 have proposed 5PP-InsP5 and InsP8 as critical regulators in phosphate export. In the opinion of the authors, are both inositol pyrophosphates relevant? What has the current study contributed to this mechanism? The manuscript confirmed the necessity of the SPX domain on XPR1 to observe the effects of SC-919, but are there certain conclusions that can be drawn from the use of this new pharmacological tool? How could the pharmacological tool be used in combination with genetics to address the specificity issues in PP-InsP signaling? In my opinion, there is a lot of potential, and the authors should discuss this more carefully.

Response,

Thank you for your comment. Our data suggest that IP6K has a role in regulating phosphate *in vivo*. However, owing to the limitation of having only an IP6K inhibitor, we cannot confirm whether 5PP-InsP5 or InsP8 are equally relevant (pages 11-12, lines 244-253). To elucidate this possibility, we must include a specific PPIP5Ks inhibitor, which is a subject for another study. Shears et al. showed that InsP8 likely to have a dominant role in regulating phosphate

in cells. IP6K inhibition decreases InsP7 and InsP8 levels, *in vivo*; thus, the reduction in circulating phosphate levels following IP6K inhibition may be mediated by a decrease in InsP8 via *in vivo* InsP7 reduction as described in the revised manuscript (pages 11-12, lines 247-249). In addition, SC-919 contribution to the inositol pyrophosphate biology was discussed in the revised manuscript to address the specificity issues in PP-InsP signaling (page 13, lines 285-290).

Minor comments:

- Line 157 and 158. IP6K instead of IPK6.

Response,

Thank you for your suggestion. This was corrected in the revised manuscript.

- Line 213 comma after IP6K

Response,

Thank you for your suggestion. This was corrected in the revised manuscript.

- Supporting Information Page 8: Synthesis of InsP7: I couldn't find the reference for the synthesis

Response,

Thank you for your suggestion. This was corrected in the revised manuscript (page 18, lines 397-398).

REVIEWER COMMENTS

Reviewer #2 (Remarks to the Author):

Authors have addressed previous comments sufficiently, i recommend accepting the manuscript

Reviewer #3 (Remarks to the Author):

The authors responded satisfactory to most of my comments, except the last one (comment 11).

Still remains the question as to whether the therapeutic effect of SC-919 on hyperphosphatemia is mainly driven by (i) (as the authors state) an altered cellular phosphate metabolism or (ii) a better kidney function.

Measurement of ATP levels in other tissues should at least give partial explanation for this. I see no reason why ATP in other tissues cannot be measured.

Reviewer #4 (Remarks to the Author):

The authors submitted a greatly improved manuscript with regards to many major comments. Some minor aspects, however, still need to be added or changed:

The authors somewhat misunderstood my previous comment about other IP6K inhibitors. I was just asking them to mention currently available inhibitors and their benefits and drawbacks. While the authors now included TNP in their IC50 determination, they still don't provide much background on TNP, as well as other IP6K inhibitors that were recently developed. Papers such as Liao, G. et al., ACS Pharmacology & Translational Science, 2021 and Wormald, M., M., et al., Bioorg Med Chem Lett, 2019 should be cited.

The IC50 values provided in Table 1 are lower than half the enzyme concentrations used, which is technically not possible. It makes sense to note them as "< value" but the enzyme concentrations need to be adapted.

I understand that the purpose of this manuscript was to address the physiological role of inositol pyrophosphates on phosphate homeostasis. To do so, however, the authors rely on a single pharmacological tool, SC-919. Consequently, the characterization of this inhibitor needs to be sufficient, so that the reader understands which conclusions can really be drawn. I appreciated the added chemical characterization of SC-919. An open question remains what the off-target activities of SC-919 against other small molecule kinases - especially IPMK, IP3K, and ITPK – are. While this may be technically difficult for the authors at this time, this open question should at least be acknowledged in the text.

Reviewer #5 (Remarks to the Author):

In this revised manuscript, the authors have attempted to address major concerns regarding in vivo actions of SC-919. First of all, by measuring tissue levels of SC-919 following its oral administration, authors showed high levels of SC-919 in the kidney (Table S5). With respect to the urine excretion of phosphate, authors speculated decreased plasma phosphate concentrations as a possible factor for the decreased urine phosphate excretion in SC-919-treated rats. Their interpretation should be clearly provided in the Results or Discussion. In addition, authors proposed a clamp study for further studies to fully delineate urine excretion phenotypes. Such discussion should be also described in the text to help readers in the field. Other issues related to LC/MS-based IP detection as well as revised Introduction have been nicely handled.

REVIEWER COMMENTS

Our responses to the reviewers' comments

Reviewer #2 (Remarks to the Author):

Authors have addressed previous comments sufficiently, i recommend accepting the manuscript

Response:

Thank you for the effort expended in reviewing our manuscript.

Reviewer #3 (Remarks to the Author):

The authors responded satisfactory to most of my comments, except the last one (comment 11).

Still remains the question as to whether the therapeutic effect of SC-919 on hyperphosphatemia is mainly driven by (i) (as the authors state) an altered cellular phosphate metabolism or (ii) a better kidney function.

Measurement of ATP levels in other tissues should at least give partial explanation for this. I see no reason why ATP in other tissues cannot be measured.

Response:

Thank you for your comment. Considering the severe kidney disease condition, altered cellular phosphate is likely to be a primary driver of SC-919 in lowering phosphate in adenine rats with hyperphosphataemia. In fact, as shown in Fig. 4 e and f, SC-919 is still active in improving hyperphosphataemia in bilaterally nephrectomised rats. This strongly indicates that altered cellular phosphate metabolism outside the kidney contributes to lowered plasma phosphate levels in conditions such as hyperphosphataemia. We share our preliminary data showing that SC-919 increases ATP levels by approximately 1.2-fold in the muscles and liver. Considering the organ volume percentage in the body (see Table below), we speculate that the muscles and other organs play a role in

improving hyperphosphataemia when IP6K is inhibited. In the future, we will aim to answer this important question through a well-designed study. However, SC-919-induced improvement of kidney function may regulate plasma phosphate levels better. This possibly results in the improvement of hyperphosphataemia, at least in part. This explanation has been included in the Discussion (page 13, line 278).

Table 1. Organ volume (mL) for rat and Rhesus monkey

	Rat 0.277 kg	% of total volume	Monkey 4.4 kg	% of total volume
Liver	11.17	4.0	140.31	3.2
Kidneys	2.49	0.9	21.5	0.5
Brain	11.17	4.0	90.3	2.1
Bone	17.13	6.2	400	9.1
Fat	10.84	3.9	500	11.4
Guts	10.84	3.9	156.99	3.6
Pericardium	0.87	0.3	21.51	0.5
Lungs	1.08	0.4	45.49	1.0
Muscle	132.28	47.8	2000	45.4
Skin	43.37	15.7	444.1	10.1
Spleen	0.65	0.2	3.89	0.1
Thymus	0.76	0.3	1	0.0
Extra-organ blood	8.71	3.1	393.8	8.9
Total volume	276.96	100.0	4400.89	100.0
Total blood volume	20.7	7.5	655.8	14.9

*Modified from *J Pharm Sci.* 2012 Mar;101(3):1221-41. doi: 10.1002/jps.22811.

Modified from Hall, C., Lueshen, E., Mošat', A. and Linninger, A.A. (2012), Interspecies scaling in pharmacokinetics: A novel whole-body physiologically based modeling framework to discover drug biodistribution mechanisms in vivo. *J. Pharm. Sci.*, **101**: 1221-1241. <https://doi.org/10.1002/jps.22811>, with permission from Elsevier

Reviewer #4 (Remarks to the Author):

The authors submitted a greatly improved manuscript with regards to many major comments. Some minor aspects, however, still need to be added or changed:

The authors somewhat misunderstood my previous comment about other IP6K inhibitors. I was just asking them to mention currently available inhibitors and their benefits and drawbacks. While the authors now included TNP in their IC50 determination, they still don't provide much background on TNP, as well as other IP6K inhibitors that were recently developed. Papers such as Liao, G. et al., ACS Pharmacology & Translational Science, 2021 and Wormald, M., M., et al., Bioorg Med Chem Lett, 2019 should be cited.

Response:

Thank you for your comment and clarification. The revision now includes the background information about other IP6K inhibitors. Moreover, the reports suggested by the reviewer have been cited (page 14, lines 299-311).

The IC50 values provided in Table 1 are lower than half the enzyme concentrations used, which is technically not possible. It makes sense to note them as "< value" but the enzyme concentrations need to be adapted.

Response:

Thank you for your comment. The current method was optimised to increase the assay sensitivity for measuring inhibitory activity of SC-919 for each enzyme. As mentioned in the response for the first revision, we tried to measure the inhibitory activity more accurately. However, this may take a considerable amount of time.

I understand that the purpose of this manuscript was to address the physiological role of inositol pyrophosphates on phosphate homeostasis. To do so, however, the authors rely on a single pharmacological tool, SC-919. Consequently, the characterization of this inhibitor needs to be sufficient, so that the reader understands which conclusions can really be drawn. I appreciated the added chemical characterization of SC-919. An open question remains what the off-target activities of SC-919 against other small molecule kinases - especially IPMK, IP3K, and ITPK – are. While this may be technically difficult for the

authors at this time, this open question should at least be acknowledged in the text.

Response:

Thank you for your comment. As pointed out, the characterization of this inhibitor should be sufficient. We are confident of the potency, physicochemical properties, and *in vivo* potency of SC-919. However, the inhibitory activity of SC-919 against related kinases was not evaluated in detail in the current study; this will have to be verified in a future study. We have now included this statement in the revision (page 14, lines 304-311). We share preliminary *in vivo* data regarding the measurement of IP4 and IP5 levels *in vivo*. SC-919 did not alter these levels *in vivo*.

Reviewer #5 (Remarks to the Author):

In this revised manuscript, the authors have attempted to address major concerns regarding *in vivo* actions of SC-919. First of all, by measuring tissue levels of SC-919 following its oral administration, authors showed high levels of SC-919 in the kidney (Table S5). With respect to the urine excretion of phosphate, authors speculated decreased plasma phosphate concentrations as a possible factor for the decreased urine phosphate excretion in SC-919-treated rats. Their interpretation should be clearly provided in the Results or Discussion. In addition, authors proposed a clamp study for further studies to fully delineate urine excretion phenotypes. Such discussion should be also described in the text to help readers in the field. Other issues related to LC/MS-based IP detection as well as revised Introduction have been nicely handled.

Response:

Thank you for your comment. Our interpretation has been included in the Results (page 7, line 143). Furthermore, the inhibitory role of IP6K in kidney phosphate handling has been described in the Discussion (page 11, lines 242-246).

REVIEWERS' COMMENTS

Reviewer #3 (Remarks to the Author):

With this second review, the authors satisfactorily answered to my questions.

Reviewer #4 (Remarks to the Author):

The authors have addressed all my questions and I am happy to support the publication of this manuscript.

Reviewer #5 (Remarks to the Author):

Authors nicely handled comments in the revised manuscript. No further action is needed.

REVIEWERS' COMMENTS

Reviewer #3 (Remarks to the Author):

With this second review, the authors satisfactorily answered to my questions.

Response:

Thank you for the effort expended in reviewing our manuscript.

Reviewer #4 (Remarks to the Author):

The authors have addressed all my questions and I am happy to support the publication of this manuscript.

Response:

Thank you for the effort expended in reviewing our manuscript.

Reviewer #5 (Remarks to the Author):

Authors nicely handled comments in the revised manuscript. No further action is needed.

Response:

Thank you for the effort expended in reviewing our manuscript.